# The Blessings of Multiple Treatments and Outcomes in Treatment Effect Estimation

## Abstract

Assessing causal effects in the presence of unobserved confounding is a challenging problem. Existing studies leveraged proxy variables or multiple treatments to adjust for the confounding bias. In particular, the latter approach attributes the impact on a single outcome to multiple treatments, allowing estimating latent variables for confounding control. Nevertheless, these methods primarily focus on a single outcome, whereas in many real-world scenarios, there is greater interest in studying the effects on multiple outcomes. Besides, these outcomes are often coupled with multiple treatments. Examples include the intensive care unit (ICU), where health providers evaluate the effectiveness of therapies on multiple health indicators. To accommodate these scenarios, we consider a new setting dubbed as *multiple treatments and multiple outcomes*. We then show that parallel studies of multiple outcomes involved in this setting can assist each other in causal identification, in the sense that we can exploit other treatments and outcomes as proxies for each treatment effect under study. We proceed with a causal discovery method that can effectively identify such proxies for causal estimation. The utility of our method is demonstrated in synthetic data and sepsis disease.

## 1 Introduction

Estimating average causal effects from observed data is an important problem in many areas, such as social sciences, biological sciences, and economics. A critical challenge in estimating causal effect arises from the presence of unobserved confounders. To tackle this problem, existing works relied on additional measurements, such as instrumental variables (Pokropek, 2016), proxy variables (Miao et al., 2018), and additional treatments (Wang & Blei, 2019).

In particular, the proximal causal learning (Miao et al., 2018; Tchetgen et al., 2020; Cui et al., 2023) leverages two proxy variables - a treatment-inducing proxy and an outcome-inducing proxy - to account for unmeasured confounders. With such proxies, one can identify the causal effect (Cui et al., 2023) by estimating nuisance parameters. However, these methods required to pre-specify proxy variables, which may not be feasible in real applications. Recently, another deconfounding framework was proposed Wang & Blei (2019), by exploiting multiple treatments to estimate the confounders to adjust for the confounding bias. Based on this framework, Wang & Blei (2021); Miao et al. (2022) further associated the proxy variables with other treatments for identification.

However, in many real scenarios, we are more interested in multiple outcomes rather than a single isolated outcome. Besides, these outcomes are often coupled with multiple treatments. To illustrate, consider the Intensive Care Unit (ICU) scenario with sepsis disease (Johnson et al., 2016) as a motivating example, where healthcare providers may monitor various parameters, such as White blood cell count, Mean blood pressure, and Platelets, to assess the effectiveness of therapeutic treatments including Norepinephrine, Morphine Sulfate, and Vancomycin. Such scenarios are often encountered but were ignored in the literature. To fill in this blank, we introduce a novel setting: *multiple treatments and multiple outcomes*, where we consider the case when treatments are continuous.

Our setting is a natural extension to the *multiple treatments* setting (Wang & Blei, 2019) in the sense that the shared confounder of multiple treatments also affects multiple outcomes recorded, as evidenced by extensive examples in Sec. 3. Here, we revisit the ICU example for illustration. In this example, the unobserved confounder can refer to the health outcomes at the previous stage, which not only determine the therapy dosage but also can affect outcomes at the next stage.

Moreover, having multiple outcomes of interest can aid in the mutual identification of each other. Concretely, we show that there can always exist two admissible proxies that can guarantee the identifiability for each outcome under treatment effect study. This conclusion holds as long as there exist missing edges in the bipartite graph between treatments and outcomes, which can naturally hold since some treatments may only impact a few outcomes. Back to the ICU example, morphine may have no obvious influence on the White Blood Cell Counts (Anand et al., 2004; Degrauwe et al., 2019). Under this guarantee, we identify such proxies via hypothesis testing for causal edges between each treatment and outcome. With such identified proxies, we estimate the treatment effect with a kernel-based proximal doubly robust estimator that can well handle continuous treatment effects. As a contrast, multiple treatments with a single outcome (Wang & Blei, 2019) can suffer from larger biases due to the non-identifiability of latent confounders giving rise to multiple treatments. To demonstrate the utility and effectiveness of our method, we apply it both to a synthetic dataset and Medical Information Mart for Intensive Care (MIMIC III) (Johnson et al., 2016).

It is interesting to note that recently (Zhou et al., 2020) also studied the multiple outcomes with only a single treatment. However, it relied on the *no qualitative U-A interaction* assumption ($U$, $A$ resp. denote latent confounders and treatment), which may not hold when the outcome model is complex.

**Contributions.** To summarize, our contributions are:

1. We introduce a new setting that involves multiple treatments and multiple outcomes, which can accommodate many real scenarios.
2. We show that this setting facilitates causal identification under mild conditions.
3. We employ hypothesis testing via a proper discretization of treatments to identify proxies.
4. We demonstrate the utility of our methods on both synthetic and real-world data.

## 2 RELATED WORKS

**Causal Inference with Multiple Treatments.** Wang & Blei (2019) and follow-up studies (D'Amour, 2019; D'Amour, 2019; Imai & Jiang, 2019; Miao et al., 2022) considered the scenario of multiple treatments with shared confounding structure, to adjust for confounding bias using factor models. Based on this framework, Wang & Blei (2021) further leveraged the proxy variables for identification, where certain treatments themselves act as proxies for other treatments. However, this assumption may not hold when all treatments affect the outcome. In this paper, we consider a natural extension from a single outcome to multiple outcomes under the multiple treatments setting, which facilitates the identification by exploiting certain treatments and outcomes to be proxies.

**Causal Inference with Multiple Outcomes.** Multiple outcomes are common in randomized controlled trial and recent research has explored the analysis of treatment effects in such cases (Lin et al., 2000; Roy et al., 2003). Kennedy et al. (2019) propose scaled effect measures method to estimate the effects of multiple outcomes. Additionally, Yao et al. (2022) suggested leveraging data from multiple outcomes to estimate treatment effects. In a recent work, (Zhou et al., 2020) show nonparametric identifiability under the assumption of conditional independence among at least three parallel outcomes. However, this relied on the *no qualitative U-A interaction* assumption, which may not hold when the outcome model of $Y|U, A$ is complex. **In contrast**, our method leverages multiple treatments that are often coupled with outcomes recorded, which provides proxies for identification in a more flexible way.

**Proximal Causal Learning for Identifiability.** To estimate the treatment effect on a single outcome in the presence of latent confounders, Miao et al. (2018); Tchetgen et al. (2020); Cui et al. (2023) proposed to use two proxy variables of unobserved confounders for causal identification. With such proxies, one should solve for nuisance/bridge functions under completeness assumptions, which were then used in the doubly robust estimator (Cui et al., 2023) for causal estimations. However, these works pre-specified the proxies. Furthermore, their emphasis was primarily on binary treatments. In contrast, in our scenario, our method enjoys the capability to identify proxy variables through causal discovery and estimate causal effects for continuous treatments.

## 3 MULTIPLE TREATMENTS AND MULTIPLE OUTCOMES

**Problem setup & Notations.** We consider the setting of causal inference with *multiple treatments and multiple outcomes*. Specifically, our data is composed of $n$ i.i.d samples $\{\mathbf{a}^i, \mathbf{x}^i, \mathbf{y}^i\}_{i=1:n}$ with

covariates $\mathbf{X}$, $I$ ($I > 1$) treatments $\mathbf{A} = [A_1, \cdots, A_I]$, $J$ ($J > 1$) outcomes $\mathbf{Y} := [Y_1, \cdots, Y_J]$ that recorded after treatments are received. Here, we assume that all treatments are continuous. We denote $[m] := \{1, \cdots, m\}$ for any integer $m > 0$. For a subset $S \subseteq [m]$, we denote $\mathbf{O}_S := \{O_i | i \in S\}$ as the subset of $\mathbf{O}$ for any variables $O : \Omega \to \mathbf{R}^m$ that can denote $\mathbf{A}$, $\mathbf{X}$, and $\mathbf{Y}$ in our setting. Correspondingly, we denote $\mathbf{O}_{-S} := \{O_i | i \notin S\}$ as the complementary set of $\mathbf{O}_S$. Besides, we respectively use $\mathbb{E}[\cdot]$ and $\mathbb{P}(\cdot)$ to denote the expectation and the probability distribution of a random variable. For any discrete variables $X$ and $Y$ with $K$ and $L$ categories, we define $\mathcal{P}(X|y) := [\mathbb{P}(x_1|y), \cdots, \mathbb{P}(x_K|y)]^\top$, $\mathcal{P}(x|Y) := [\mathbb{P}(x|y_1), \cdots, \mathbb{P}(x|y_l)]$ and $\mathcal{P}(X|Y) := [\mathcal{P}(X|y_1), \cdots, \mathcal{P}(X|y_L)]$ as probability matrices.

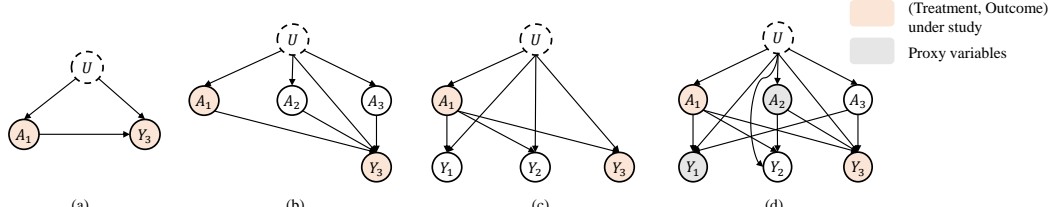

Figure 1: Different settings of causal inference: (a). single treatment and single outcome; (b) multiple treatments and single outcome; (c) single treatment and multiple outcomes; (d) ours with multiple treatments and multiple outcomes. $\mathbf{A}$, $\mathbf{Y}$, $\mathbf{U}$ respectively denote treatments, outcomes, and the unobserved confounder marked by the dotted circle. We mark the pair of (treatment, outcome) involved in the causal effect study as blue and corresponding proxy variables as gray.

Our goal is to estimate the average treatment effect (ATE) for multiple treatments of interest. Specifically, for each outcome $Y_j$ we would like to estimate:

$$\mathbb{E}[Y_j | \mathrm{do}(\mathbf{a}_\mathcal{S})], \tag{1}$$

$\forall \mathbf{A}_S \subset \mathbf{A}$. Note that our setting in Fig. 1 (d) is a natural extension from the *multiple treatments* setting (Wang & Blei, 2019) in Fig. 1 (b), from a single outcome to multiple outcomes. In many real-world scenarios, instead of a single experiment, it is more interesting to evaluate the effectiveness of treatments/interventions to multiple outcomes (Yao et al., 2022) of interest. To illustrate, consider the Intensive Care Unit (ICU) scenario with sepsis disease (Johnson et al., 2016) as a motivating example of Fig. 1 (d), where we can not only observe multiple treatments such as Norepinephrine ($A_1$), Morphine Sulfate ($A_2$), and Vancomycin ($A_3$) but also record health indicators such as White blood cell count ($Y_1$), Mean blood pressure ($Y_2$), and Platelets ($Y_3$). As shown in our learned causal graph in Fig. 1 (d) that aligns well with existing findings, different health indicators are associated with different subsets of treatments. As each health indicator represents a crucial aspect of sepsis, our focus is on examining the treatment effects associated with each of them. Another example is the recommendation system (Yao et al., 2022), where companies conducted A/B experiments to test the effectiveness of website layout changes towards multiple metrics including user retention, and server CPU usage. Note that Zhou et al. (2020) also considered the multiple outcomes setting, but it was only with a single treatment, as shown in Fig. 1 (c). This setting may not capture the scenarios where multiple treatments together are under study, such as the ICU scenario mentioned earlier.

The additional incorporation of multiple outcomes is not only of practical interest but can also facilitate the identification of causal effect in Eq. 1 when the ignorability condition fails. This is because, for each pair of $(\mathbf{A}_S, Y_j)$ under study, we can leverage other treatments and outcomes as proxy variables that can identify the causal effect under unobserveness (Miao et al., 2018). As shown in Fig. 1, we can respectively use $A_2$ and $Y_1$ as outcome-induced and treatment-induced proxies that are sufficient to identify the causal effect of $A_1$ to $Y_3$. In contrast, the *multiple treatments* setting (Fig. 1 (b)) (Wang & Blei, 2021; 2019) required the existence of a null proxy that has no effect on the outcome, which may not always be satisfied for a fixed outcome. On the other hand, the *multiple outcomes* setting (Fig. 1 (c)) relied on the *no qualitative U-A interaction* assumption, which may not hold when the outcome model of $Y|U, A$ is complex.

We first introduce some basic assumptions that our method to identify Eq. 1 is built upon.

**Assumption 3.1** (Structural Causal Model with Shared Confounding). *We assume a structural causal model (SCM) (Pearl, 2009) over $\{\mathbf{U}, \mathbf{A}, \mathbf{X}, \mathbf{Y}\}$, where we assume that (i) $\mathbf{U}$ denotes unobserved confounders that shared among all treatments and outcomes: $Y_j(\mathbf{a}) \perp \mathbf{A}|\mathbf{X}, \mathbf{U}$ for each*

$j$, where $Y_j(\mathbf{a})$ denotes the potential outcome with $\mathbf{A} = \mathbf{a}$ received; **(ii)** there are no directed edges between treatments: $A_{i_1} \perp A_{i_2} | \mathbf{X}, \mathbf{U}$ for any $i_1 \neq i_2$, or between outcomes: $Y_{j_1}(\mathbf{a}) \perp Y_{j_2}(\mathbf{a}) | \mathbf{X}, \mathbf{U}$ for any $j_1 \neq j_2$ and $\mathbf{a}$.

Assumption 3.1 **(i)** is similar to and naturally remains valid as long as the shared confounding assumption among multiple treatments (Wang & Blei, 2019) holds. This is because the unobserved confounder $U$ often represents underlying factors that can impact various interconnected outcomes. To illustrate, consider the examples in Wang & Blei (2019).

- **Genome-wide association studies (GWAS)**. The goal is to evaluate the effect of genes on a specific trait, where $\mathbf{U}$ refers to shared ancestry that can also affect other related traits.

- **Computational neuroscience**. The causes are multiple measurements of the brain's activity. The effect is a measured behavior. $\mathbf{U}$ refers to dependencies among neural activity, which can affect multiple behaviors and thoughts.

- **Social Science**. The policymakers evaluated the effect of social programs on social outcomes that include poverty levels and upward mobility (Morgan & Winship, 2015), which can be simultaneously affected by the program preferences, *i.e.*, $\mathbf{U}$.

- **Medicine**. The confounder $\mathbf{U}$ refers to treatment preferences, which can simultaneously affect health outcomes/indicators that are linked to those treatments.

Note that in some other cases (*e.g.*, the sepsis disease in the ICU), $\mathbf{U}$ can include the outcomes recorded at the last step, which can also affect the treatments and outcomes at the next step.

In addition, the absence of direct causal relationships between outcomes or between treatments, as stipulated in assumption 3.1 **(ii)**, naturally applies to many scenarios. These scenarios include the aforementioned ones in which outcomes are only affected by treatments and covariates while treatments are only affected by covariates. Additionally, it encompasses other examples where treatments can be additionally affected by outcomes recorded at the last step.

Next, we introduce the *null-proxy* assumption that guarantees the identification of causal effects by exploiting other treatments and outcomes.

**Assumption 3.2** (Null-proxy for $\mathbf{A}_S$ and $Y_j$). *We assume that at least one of the following holds: (i) $|\mathbf{Y}| \geq 3$ and there is at least one missing edge from $\mathbf{A}_S$ to $\mathbf{Y}_{-j}$; (ii) $|\mathbf{A}_{-S}| \geq 2$ and there exists at least one missing edge from $\mathbf{A}_{-S}$ to $\mathbf{Y}_j$; (iii) there is at least one missing edge from $\mathbf{A}_{-S}$ to $\mathbf{Y}_{-j}$.*

*Remark* 3.3. Note that If we are interested in single treatment effect, the above assumption holds for all $i \in [I]$ and $j \in [J]$ as long as **(i)** and **(ii)** are respectively required in Zhou et al. (2020) and Wang & Blei (2021). If these conditions fail, we can also identify the ATE as long as there are at most $IJ - 2$ edges in the bipartite graph between $\mathbf{A}$ and $\mathbf{Y}$.

*Remark* 3.4. We will show later that this assumption can be empirically tested from data.

Roughly speaking, this assumption implies the presence of missing edges from $\mathbf{A}$ and $\mathbf{Y}$, as long as the bipartite graph between them is not overly dense. It can naturally hold since in scenarios with multiple treatments and multiple outcomes, some outcomes may be only associated with a few treatments. To illustrate, consider the GWAS example in which each trait can be associated with only a few genes, and Fig. 1 (d) in the example of sepsis disease in ICU where White blood cell count ($Y_1$) is only associated with Norepinephrine ($A_1$) and Vancomycin ($A_3$). To understand how this assumption affects the ATE in Eq. 1, it is important to note that two variables within the $\mathbf{A}_{-S} \cup \mathbf{Y}_{-j}$, characterized by the presence of missing edges, can serve as valid proxies for Eq. 1 to be identifiable. Compared to Wang & Blei (2021) that assumed the existence of null proxies for a single outcome, our assumption is easier to satisfy by exploiting other treatments and outcomes provided as proxies. Back to the example of sepsis disease in Fig. 1 (d), the null proxy does not exist (as illustrated in Fig. 1 (b)) if we only consider Platelets ($Y_3$) as a single outcome; while we can take $A_2$ and $Y_1$ as proxies since $A_2$ does not affect $Y_1$. Recently, an alternative null treatment approach was introduced (Miao et al., 2022). However, it required at least half of the treatments do not causally affect the outcome, which is much more stringent than ours.

To guarantee identifiability for each outcome, we additionally require *positivity* assumption that was commonly made in proximal causal inference (Miao et al., 2018; Cui et al., 2023).

**Assumption 3.5** (Consistency and Positivity). *(i) $Y^{(A,Z)} = Y$, (ii) $0 < p(A = a|U, X) < 1$ a.s.*

## 4 IDENTIFIBILITY

We show that Eq. 1 is identifiable based on proximal causal learning. Different from previous works that pre-specify proxies, we propose to select admissible proxies based on causal discovery.

### 4.1 IDENTIFICATION WITH PROXIES

We first show that as long as assump. 3.2 holds, there always exist two admissible proxies namely $W_{S,j}, Z_{S,j}$ that can help to identify Eq. 1. According to Miao et al. (2018), such two proxies should adhere to the following conditional independence:

$$Z_{S,j} \perp Y_j \mid \mathbf{A}_S, \mathbf{A}_{-S} \backslash W_{S,j}, \mathbf{U}, \tag{2}$$

$$(Z_{S,j}, A_S) \perp W_{S,j} \mid \mathbf{A}_{-S} \backslash W_{S,j}, \mathbf{U}, \tag{3}$$

as illustrated in Fig. 1 (d) by taking $S := \{1\}, j := 3, W_{S,j} = A_2$ and $Z_{S,j} = Y_1$. Note that here we have omitted $\mathbf{X}$ in Eq. 2, 3 since they are observed confounding variables and can thus be easily adjusted for. Such conditions imply that there is no directed edge between $Z_{S,j}$ and $W_{S,j} \cup Y_j$. In fact, these conditions can be guaranteed by assump. 3.2, as shown in the following lemma.

**Lemma 4.1.** *Suppose Assumptions 3.1 and 3.2 hold for $\mathbf{A}_S \subset \mathbf{A}$ and $Y_j \in \mathbf{Y}$. Then, there exist two admissible proxies $(Z_{S,j}, W_{S,j})$ that satisfy Eq. 2, 3.*

*Remark* 4.2. Intuitively, Lemma. 4.1 is guaranteed by the condition of the missing directed edges, *i.e.*, assump. 3.2. Specifically, if $J \geq 3$, any two $Y_{j_1}, Y_{j_2} \notin \mathbf{Y}_{-j}$ can be taken as $Z_{S,j}$ and $W_{S,j}$ since we have assumed that outcomes do not interact with each other. Otherwise, when there exists missing edges from $\mathbf{A}_S$ to $\mathbf{Y}$, we can also find $(Z_{S,j}, W_{S,j})$. Please refer to the proofs for details.

This condition means we can find proxies from other treatments and outcomes. We will show in the next section how to identify proxies $Z_{S,j}$ and $W_{S,j}$ via causal discovery. Finally, we need the completeness assumptions for two bridge functions for causal identification.

**Assumption 4.3.** *Let $\nu$ denote any square-integrable function. Then for any $S$ and $j \in J$, we have*

1. *(Completeness for outcome bridge functions). We assume that $\mathbb{E}[\nu(\mathbf{U})|\mathbf{a}_S, W_{S,j}, \mathbf{x}] = 0$ and $\mathbb{E}[\nu(Z_{S,j})|\mathbf{a}_S, W_{S,j}, \mathbf{x}] = 0$ for all $(\mathbf{a}_S, \mathbf{x})$ iff $\nu(\mathbf{U}) = 0$ almost surely.*
2. *(Completeness for treatment bridge functions). We assume that $\mathbb{E}[\nu(\mathbf{U})|\mathbf{a}_S, Z_{S,j}, \mathbf{x}] = 0$ and $\mathbb{E}[\nu(W_{S,j})|\mathbf{a}_S, Z_{S,j}, \mathbf{x}] = 0$ for all $(\mathbf{a}_S, \mathbf{x})$ iff $\nu(\mathbf{U}) = 0$ almost surely.*

Assump. 4.3, which has been widely adopted in the literature of causal inference (Miao et al., 2018; Cui et al., 2023; Tchetgen et al., 2020), is necessary to guarantee the existence and the uniqueness of solutions to integral equations. Similar to Cui et al. (2023), we also need regularity conditions that are left in the appendix. Under these assumptions, we have the following identifiability result:

**Theorem 4.4.** *Suppose assump. 3.1-3.5, 4.3 hold for $\mathbf{A}_S$ and $Y_j$. Then $\mathbb{E}[Y_j|\mathrm{do}(\mathbf{a}_S)]$ is identifiable.*

In contrast to previous works such as Miao et al. (2018); Wang & Blei (2021); Miao et al. (2022) that required the pre-specification of proxy variables, our method can identify the proxies by exploiting multiple treatments and multiple outcomes, which can be easily obtained in many real scenarios. Particularly, even when the condition of *null proxy treatment* fails, Thm. 4.4 suggests that we can also identify the causal effect by exploiting multiple outcomes. Additionally, thanks to multiple treatments, our assumption is weaker than that of Zhou et al. (2020).

In the next section, we show assump. 3.2 can be tested and we can find proxies via causal discovery.

### 4.2 IDENTIFY PROXIES VIA CAUSAL DISCOVERY

Assump. 4.1 and Thm. 4.4 show the existence of suitable proxies. In this section, we perform a hypothesis-testing approach to identify such proxies $W_{S,j}$ and $Z_{S,j}$ by learning the causal relations from $\mathbf{A}$ to $\mathbf{Y}$, building upon previous works by Miao et al. (2018); Liu et al. (2023b;a). With this approach, we can also test whether assump. 4.1 holds. We denote $W_{S,j}, Z_{S,j}$ as $W, Z$ for brevity.

Since we assume that the treatments do not interact with each other and the outcomes do not interact with each other, we can conduct the null hypothesis: $\mathbb{H}_0 : A_i \perp Y_j|\mathbf{U}$, for each $A_i \in \mathbf{A}$ and $Y_j \in \mathbf{Y}$. Since treatments do not interact with each other, rejecting the null hypothesis indicates a direct causal edge from $A_i$ to $Y_j$. Similar to estimating ATE, previous works (Miao et al., 2018) also required the prior specification of one proxy to carry out such hypothesis testing on discrete

treatments. In our paper, we extend to continuous treatments; moreover, our method only requires one single proxy that could be taken from any $A_{i'} \neq A_i$. Next, we introduce some fundamental assumptions that are easy to hold in many scenarios.

**Assumption 4.5** (NULL-TV Lipschitzness). *For any $i \in [I]$, we assume $\mathbf{u} \mapsto \mathbb{P}(A_{i'}|A_i, \mathbf{U} = \mathbf{u})$ and $\mathbf{u} \mapsto \mathbb{P}(A_{i'}|\mathbf{U} = \mathbf{u})$ are Lipschitz continuous with respect to Total Variation (TV) distance. Mathematically speaking, $\forall \mathbf{u}, \mathbf{u}' \in \mathrm{supp}(\mathbf{U})$, we have*

$$\mathrm{TV}\left(\mathbb{P}(A_{i'}|A_i, \mathbf{U} = \mathbf{u}), \mathbb{P}(A_{i'}|A_i, \mathbf{U} = \mathbf{u}')\right) \leq L_{A_{i'}}|\mathbf{u} - \mathbf{u}'|;$$
$$\mathrm{TV}\left(\mathbb{P}(A_{i'} \mid \mathbf{U} = \mathbf{u}), \mathbb{P}(A_{i'}|\mathbf{U} = \mathbf{u}')\right) \leq L_{A_{i'}|A_i}|\mathbf{u} - \mathbf{u}'|.$$

This assumption is widely used in conditional independence testing or conditional density estimation (Warren, 2021; Neykov et al., 2021; Li et al., 2022), and has recently been employed in causal inference. Generally speaking, it restricts our attention to scenarios where the conditional distributions are appropriately smooth, which can hold in many scenarios such as *Additive Noise Model* (ANM), Huber contamination model, exponential tilting model, *etc* [1]. With such a smoothness, assump. 4.5 allows us to obtain conditional independence even after the discretizing continuous treatments. Specifically, we denote $N$ bins $\{\mathcal{U}_n\}_{n=1}^N$ and $\{\mathcal{A}_n\}_{n=1}^N$ as measurable partition of $\mathbf{U}$ and $A_{i'}$ in the hypothesis testing of $\mathbb{H}_0 : A_i \perp Y_j|\mathbf{U}$. Denote $\overline{\mathbf{U}}$ and $\overline{A}_{i'}$ as discretized version of $\mathbf{U}$ and $A_{i'}$ such that $\overline{\mathbf{U}} = k$ iff $\mathbf{U} \in \mathcal{U}_k$. Then we have

$$\mathbb{P}(a_{i'} \in \mathcal{A}_n|a_i) = \sum_{k=1}^N \mathbb{P}(a_{i'} \in \mathcal{A}_n|\mathbf{U} \in \mathcal{U}_k, a_i)\mathbb{P}(\mathbf{U} \in \mathcal{U}_k|a_i) := \mathcal{P}(\mathbf{a}_{i'} \in \mathcal{A}_n|\overline{\mathbf{U}}, a_i)\mathcal{P}(\overline{\mathbf{U}}|a_i),$$

$$\mathbb{P}(Y_j \leq y|a_i) = \sum_{k=1}^N \mathbb{P}(Y_j \leq y|\mathbf{U} \in \mathcal{U}_k, a_i)\mathbb{P}(\mathbf{U} \in \mathcal{U}_k|a_i) := \mathcal{P}(Y_j \leq y|\overline{\mathbf{U}}, a_i)\mathcal{P}(\overline{\mathbf{U}}|a_i).$$

When $N$ is large enough, we have $\mathbb{P}(a_{i'} \in \mathcal{A}_n|\mathbf{U} \in \mathcal{U}_k, a_i) \overset{(1)}{\approx} \mathbb{P}(a_{i'} \in \mathcal{A}_n|\mathbf{U} = u, a_i) \overset{(2)}{=} \mathbb{P}(a_{i'} \in \mathcal{A}_n|\mathbf{U} = u) \overset{(3)}{\approx} \mathbb{P}(a_{i'} \in \mathcal{A}_n|\mathbf{U} \in \mathcal{U}_k)$ for any $u \in \mathcal{U}_k$, where "(2)" is due to conditional independence between $A_{i'}$ and $A_i$. To see "(1)" and "(3)", we note that for $\mathbb{P}(a_{i'} \in \mathcal{A}_n|\mathbf{U} \in \mathcal{U}_k, a_i)$,

$$\mathrm{TV}\left(\mathbb{P}(a_{i'} \in \mathcal{A}_n|\mathbf{U} \in \mathcal{U}_k, a_i), \mathbb{P}(a_{i'} \in \mathcal{A}_n|\mathbf{U} = u, a_i)\right)$$
$$= \mathrm{TV}\left(\frac{1}{\mathbb{P}(\mathbf{U} \in \mathcal{U}_k|a_i)}\int_{u' \in \mathcal{U}_k}\mathbb{P}(a_{i'} \in \mathcal{A}_n|u', a_i)\mathrm{d}\mathbb{P}(u'|a_i), \mathbb{P}(a_{i'} \in \mathcal{A}_n|\mathbf{U} = u, a_i)\right)$$
$$= \int_{\mathcal{U}_k}\mathrm{TV}\left(\mathbb{P}(a_{i'} \in \mathcal{A}_n|\mathbf{U} = u', a_i), \mathbb{P}(a_{i'} \in \mathcal{A}_n|\mathbf{U} = u, a_i)\right)\mathrm{d}\left(\frac{\mathbb{P}(u'|a_i)}{\mathbb{P}(\mathcal{U}_k|a_i)}\right) \leq L|u' - u| \leq \varepsilon,$$

when $\mathrm{diam}(\mathcal{U}_k) \leq \frac{\varepsilon}{L}$ that can be satisfied as long as $N$ is large enough. We then have $\mathcal{P}(\overline{A}_{i'}|\overline{\mathbf{U}}) \approx \mathcal{P}(\overline{A}_{i'}|\overline{\mathbf{U}}, a_i)$. Similarly, we have $\mathbb{P}(Y_j \leq y|\overline{\mathbf{U}}, a_i) \approx \mathbb{P}(Y_j \leq y|\overline{\mathbf{U}})$ when $\mathbb{H}_0$ holds. That implies conditional independence $Y_j \perp A_i|\mathbf{U} \in \mathcal{U}_k$ for each $k$, which is crucial for hypothesis testing.

Similar to Miao et al. (2018); Liu et al. (2023b), for hypothesis testing, we assume that

**Assumption 4.6.** *For each $a_i \in \mathrm{supp}(A_i)$, the matrix $\mathcal{P}(\overline{A}_{i'}|\overline{\mathbf{U}}, a_i)$ is invertible.*

Under assump. 4.6, we have $\mathcal{P}(\overline{\mathbf{U}}|a_i) \approx \mathcal{P}(\overline{A}_{i'}|\overline{\mathbf{U}})^{-1}\mathcal{P}(\overline{A}_{i'}|a_i)$ and that

$$\mathbb{P}(Y_j \leq y|a_i) \approx \mathcal{P}(Y_j \leq y|\overline{\mathbf{U}}, a_i)\mathcal{P}(A_{i'}|\overline{\mathbf{U}})^{-1}\mathcal{P}(A_{i'}|a_i).$$

As mentioned earlier, if $\mathbb{H}_0$ holds, $\mathbb{P}(Y_j \leq y|\overline{\mathbf{U}}, a_i) \approx \mathbb{P}(Y_j \leq y|\overline{\mathbf{U}})$. In this regard, $\mathcal{P}(\overline{A}_{i'}|a_i)$ is the only source of variability with respect to $a_i$. Similar to Miao et al. (2018), we can therefore determine whether $\mathbb{H}_0$ holds by testing the linearity between $\mathbb{P}(Y_j \leq y|a_i)$ and $\mathcal{P}(\overline{A}_{i'}|a_i)$. To test such a linearity, we should make sure that $\mathbb{P}(Y_j \leq y|a_i)$ and $\mathcal{P}(\overline{A}_{i'}|a_i)$ can be estimated well, as shown in the following assumption that can easily hold via Maximum Likelihood Estimation (MLE).

**Assumption 4.7.** *Denote $q_y := \{\mathbb{P}(Y_j \leq y|a_i)\}_{i=1}^M$ and $Q := \{\mathcal{P}(\overline{A}_{i'}|a_i)\}_{i=1}^M$. Suppose we have available estimators $(\widehat{q}_y, \widehat{Q})$ that satisfy*

$$\sqrt{n}(\widehat{q}_y - q_y) \xrightarrow{d} N(0, \Sigma_y) \quad \widehat{Q} \xrightarrow{p} Q, \widehat{\Sigma}_y \xrightarrow{p} \Sigma_y, \text{ where } \widehat{\Sigma}_y \text{ and } \Sigma_y \text{ are positive-definite.} \quad (4)$$

---

[1] Please refer to the appendix for more examples that satisfy the TV Lipschitzness.

Under assump. 4.7, we can calculate the residue $\xi_y$ of regressing $\widehat{\Sigma}_y^{-1/2}\widehat{q}_y$ on $\widehat{\Sigma}_y^{-1/2}\widehat{Q}$, and check how far $\xi_y$ is away from 0 to assess whether $\mathbb{H}_0$ is correct. Formally, we have the following result.

**Theorem 4.8.** *Denote $Q_1 := \Sigma_y^{-1/2}Q^\top$, $\Omega_y := I - Q_1\left(Q_1^\top Q_1\right)^{-1}Q_1^\top$. We thus have $\xi_y = \widehat{\Omega}_y\widehat{Q}_1$. We select $a_1, ..., a_M$ with $M > N$. Then under assump. 4.5-4.7, if $\mathbb{H}_0$ is correct, we have $\sqrt{n}\xi_y \xrightarrow{d} N(0, \Omega_y)$, where the rank of $\Omega_y$ is $M - N$ and $n\xi_y^\top\xi_y \to \chi^2_{M-N}$.*

Given a significance level $\alpha$, one can reject $\mathbb{H}_0$ as long as $n\xi_y^\top\xi_y$ exceeds the $(1-\alpha)$th quantile of $\chi^2_{M-N}$, which guarantees a type I error no larger than $\alpha$ asymptotically. Thm. 4.8 demonstrates that under certain conditions, the detection of conditional independence is feasible with only a single proxy variable $A_{i'}$ for $\mathbf{U}$. With Thm. 4.8, we can conduct hypothesis testing iteratively for each edge between $\mathbf{A}$ and $\mathbf{Y}$. In this regard, we can test assump. 3.2 and find proxies $W_{S,j}$ and $Z_{S,j}$ that satisfy conditional independence in Eq. 2, 3. Once we have identified such proxies, we can perform a causal estimation of ATE, which will be introduced in the next section.

## 5 ESTIMATION

With identified proxies $W_{S,j}$ and $Z_{S,j}$ at hand, we introduce our method to estimate the causal effect. Following Tchetgen et al. (2020); Cui et al. (2023), we first solve the outcome bridge functions $h$ and treatment bridge functions $q$ from the following Fredholm integral equations, where we replace $W_{S,j}$ and $Z_{S,j}$ with $W$ and $Z$ for brevity:

$$\mathbb{E}[Y - h(A, W)|A, Z] = 0, \quad \mathbb{E}\left[q(A, Z) - 1/p(A|W)|A, W\right] = 0. \tag{5}$$

Intuitively, the bridge function $h$ and $q$ respectively serve a similar role as the regression function $\mathbb{E}[Y|a, W]$ and the propensity score $1/p(a|Z)$ in classical causal inference. To solve $h, q$, we use the Maximum Moment Restriction (Mastouri et al., 2021; Xu et al., 2021) with kernel method:

$$\min_{h \in \mathcal{H}_{\mathcal{AW}}} \frac{1}{n^2} \sum_{i,j=1}^{n} (y_i - h_i)(y_j - h_j)k_{ij}^g + \lambda_h \|h\|_{\mathcal{H}_{\mathcal{AW}}}^2 \tag{6}$$

$$\min_{h \in \mathcal{Q}_{\mathcal{AZ}}} \frac{1}{n^2} \sum_{i,j=1}^{n} (1/p_i - q_i)(1/p_j - q_j)k_{ij}^m + \lambda_q \|q\|_{\mathcal{Q}_{\mathcal{AZ}}}^2, \tag{7}$$

where $p_i := p(a_i|w_i)$ denotes the propensity score, and $h$ and $q$ belong to the reproducing kernel Hilbert space (RKHS) $\mathcal{H}_{\mathcal{AW}}, \mathcal{H}_{\mathcal{AZ}}$ with $\|\cdot\|_{\mathcal{H}_{\mathcal{AW}}}, \|\cdot\|_{\mathcal{H}_{\mathcal{AZ}}}$ and kernels $k_{ij}^g := k^g((a_i, z_i), (a_j, z_j))$, $k_{ij}^m = k^m((a_i, w_i), (a_j, w_j))$. After estimating $h$ and $q$, we employ Colangelo & Lee (2020); Wu et al. (2023) to estimate the causal effect:

$$\mathbb{E}[Y| \operatorname{do}(a)] \approx \mathbb{E}_n[K_{h_{\text{bw}}}(A - a)q(a, Z)(Y - h(a, W)) + h(a, W))], \tag{8}$$

where the indicator function $\mathbb{I}(A = a)$ in the doubly robust estimator for binary treatments (Colangelo & Lee, 2020) is replaced with the kernel function $K_{h_{\text{bw}}}(a_i - a) = 1/h_{\text{bw}}K((a_i - a)/h_{\text{bw}})$ ($h_{\text{bw}} > 0$ is the bandwidth), as a smooth approximation to make the estimation for continuous treatments feasible. In Wu et al. (2023), this estimator coupled with Eq. 6 was shown to enjoy the optimal convergence rate with $h_{\text{bw}} = O(n^{-1/5})$. Please refer to the appendix for details.

## 6 EXPERIMENTS

In this section, we evaluate our method on synthetic data and a real-world application that studies the treatment effect for sepsis disease.

**Compared baselines. (i) Generalized Propensity Score (GPS)** (Scharfstein et al., 1999) that estimated the causal effect with generalized propensity score; **(ii) Targeted Maximum Likelihood Estimation (TMLE)** (Van Der Laan & Rubin, 2006) that used Targeted Maximum Likelihood Estimation to estimate causal effects; **(iii) POP** (Zhou et al., 2020) that used multiple outcomes and linear structural equations to estimate causal effects; **iv) Deconf. (Linear)** (Wang & Blei, 2019) that estimated unobserved confounders from multiple treatments to estimate causal effects, where the outcome model is linear; **v) Deconf. (Kernel)** that the outcome model is kernel regression; **vi) P-Deconf. (Linear)** (Wang & Blei, 2021) that used null proxy on the basis of Wang & Blei (2019)

to estimate causal effects, where the outcome model is linear; **vii) P-Deconf. (Kernel)** that uses kernel regression to estimate the outcome model.

**Evaluation metrics.** We calculate the causal Mean Absolute Error (cMAE) across 10 equally spaced points in $\mathrm{supp}(A) := [0, 1]$: $\mathrm{cMAE} := \frac{1}{10} \sum_{i=1}^{10} |\mathbb{E}[Y^{a_i}] - \hat{\mathbb{E}}[Y^{a_i}]|$. The truth $\mathbb{E}[Y^a]$ is derived through Monte Carlo simulations comprising 10,000 replicates of data generation for each $a$.

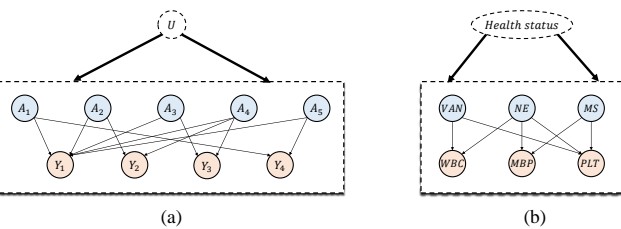

(a)  (b)

Figure 2: The causal graph over treatments (blue) and outcomes (yellow) of (a) synthetic data and (b) real-world data. The DAG of (b) is estimated using hypothesis testing in Sec. 4.2

## 6.1 SYNTHETIC STUDY

**Data generation.** We follow the DAG in Fig. 2 and following structural equations to generate data: $U \sim \mathrm{Uniform}[-1, 1]$, $A_i = g_i(U) + \epsilon_i$, with $g_i$ chosen from $\{linear, tanh, sin, cos, sigmoid\}$ and $\epsilon_i \sim N(0, 1)$, and $\mathbf{Y}$ is a non-linear structure, namely $Y_1 = 2\sin(1.4A_1 + 2A_3^2) + 0.5(A_2 + A_4^2 + A_5) + A_3^3 + U + \epsilon_1$, $Y_2 = -2\cos(1.8A_2) + 1.5A_4^2 + U + \epsilon_2$, $Y_3 = 0.7A_3^2 + 1.2A_4 + U + \epsilon_3$, and $Y_4 = 1.6e^{-A_1+1} + 2.3A_5^2 + U + \epsilon_4$ with $\{\epsilon_i\}_{i=1}^4 \sim N(0, 1)$.

**Implementation details.** For hypothesis testing in identifying proxies, we set the significant level $\alpha$ as 0.05, the $A, W, Y$ are discretized by quantile and the bin numbers of $A, W, Y$ as $I := |A| = 15, K := |W| = 8, L := |Y| = 5$, respectively. We estimate $q_y, Q$ in Eq. 4 using empirical probabilities. For the causal estimator in Eq. 8, we choose the Gaussian Kernel with bandwidth $h_{\mathrm{bw}} = 1.5\hat{\sigma}_A n^{-1/5}$ where $\hat{\sigma}_A$ is the estimated standard deviation (std) of $A$. We run each algorithm 20 times to calculate the average cMAE.

**Causal effect estimation.** We estimate the average causal effect of $A_3 \to Y_1$, $(A_1, A_3) \to Y_1$, $A_2 \to Y_2$, $(A_1, A_5) \to Y_4$ and report the results in Tab. 1. As we can see, our approach consistently provides accurate estimates of causal effects. In contrast, the GPS and TMLE suffer from large biases as the ignorability condition does not hold. Besides, Deconf. method also suffers from large biases due to the non-identifiability of latent confounders giving rise to multiple treatments. Although P-Deconf. performs well under the existence of a null proxy, it fails to estimate well when this condition is not met, as seen in the case of $A_3$ (or $A_1, A_3$) to $Y_1$ where there is no null proxy for $Y_1$. Additionally, the biases in the POP method may arise from the requirement of the condition that each treatment impacts all outcomes, which does not hold for all treatments in our scenario.

Table 1: C-MAE of our method and other baselines on synthetic data. Each method was replicated 20 times and evaluated for $A \in [0, 1]$ in each replicate.

| Method | $A_3 \to Y_1$ | | $(A_1, A_3) \to Y_1$ | | $A_2 \to Y_2$ | | $(A_1, A_5) \to Y_4$ | |
|---|---|---|---|---|---|---|---|---|
| | 600 | 1200 | 600 | 1200 | 600 | 1200 | 600 | 1200 |
| GPS | $0.97_{\pm 0.33}$ | $0.64_{\pm 0.17}$ | - | - | $0.46_{\pm 0.21}$ | $0.33_{\pm 0.14}$ | - | - |
| TMLE | $0.48_{\pm 0.16}$ | $0.43_{\pm 0.16}$ | - | - | $0.68_{\pm 0.28}$ | $0.42_{\pm 0.15}$ | - | - |
| POP | $2.64_{\pm 0.49}$ | $2.57_{\pm 0.33}$ | - | - | $2.17_{\pm 1.31}$ | $1.52_{\pm 0.65}$ | - | - |
| Deconf.(Linear) | $2.52_{\pm 0.41}$ | $2.59_{\pm 0.40}$ | $6.05_{\pm 1.58}$ | $5.81_{\pm 1.49}$ | $1.23_{\pm 0.54}$ | $1.34_{\pm 0.41}$ | $1.31_{\pm 0.35}$ | $1.16_{\pm 0.30}$ |
| Deconf.(Kernel) | $0.60_{\pm 0.08}$ | $0.48_{\pm 0.12}$ | $4.12_{\pm 0.80}$ | $3.67_{\pm 1.10}$ | $0.35_{\pm 0.20}$ | $0.38_{\pm 0.19}$ | $0.24_{\pm 0.24}$ | $0.29_{\pm 0.25}$ |
| P-Deconf.(Linear) | $2.52_{\pm 0.41}$ | $2.59_{\pm 0.40}$ | $6.05_{\pm 1.58}$ | $5.81_{\pm 1.49}$ | $1.08_{\pm 0.26}$ | $1.19_{\pm 0.18}$ | $1.81_{\pm 0.44}$ | $1.77_{\pm 0.27}$ |
| P-Deconf.(Kernel) | $0.60_{\pm 0.08}$ | $0.48_{\pm 0.12}$ | $4.12_{\pm 0.80}$ | $3.67_{\pm 1.10}$ | $\mathbf{0.26_{\pm 0.16}}$ | $\mathbf{0.21_{\pm 0.14}}$ | $0.19_{\pm 0.12}$ | $\mathbf{0.20_{\pm 0.10}}$ |
| **Ours** | $\mathbf{0.28_{\pm 0.09}}$ | $\mathbf{0.27_{\pm 0.09}}$ | $\mathbf{0.49_{\pm 0.20}}$ | $\mathbf{0.48_{\pm 0.14}}$ | $0.29_{\pm 0.14}$ | $0.23_{\pm 0.09}$ | $\mathbf{0.17_{\pm 0.09}}$ | $0.22_{\pm 0.14}$ |

**Hypothesis Testing for Causal Discovery.** To further explain the advantage over the POP that also considers multiple outcomes, we compare the accuracy of causal relations inference. Concretely, we measure the precision, recall, and the $F_1$ score of inferred causal relations with $n = 600$ samples,

and the result shows that our method is more accurate: $F_1$ (ours: $0.78 \pm 0.06$ vs POP: $0.57 \pm 0.07$), precision (ours: $0.91 \pm 0.08$ vs POP: $0.45 \pm 0.03$), and recall (ours: $0.69 \pm 0.06$ vs POP: $0.79 \pm 0.19$).

## 6.2 TREATMENT EFFECT FOR SEPSIS DISEASE

In this section, we apply our method to the treatment effect estimation for sepsis disease.

**Data Preparation.** We consider the Medical Information Mart for Intensive Care (MIMIC III) dataset (Johnson et al., 2016), which consists of electronic health records from patients in the ICU. From MIMIC III, we extract 1,165 patients with sepsis disease. For these patients, three treatment options are recorded during their stays: Vancomycin (VAN), Morphine Sulfate (MS), and Norepinephrine (NE), which are commonly used to treat sepsis patients in the ICU. After receiving treatments, we record several blood count indexes, among which we consider three outcomes in this study: White blood cell count (WBC), Mean blood pressure (MBP), and Platelets (PLT).

**Implementation details.** For hypothesis testing, we set $\alpha = 0.05$, and the bin numbers for uniform discretization of $A, W, Y$ as $I := |A| = 10, K := |W| = 6, L := |Y| = 5$, respectively. The estimation of $q_y, Q$ in Eq. 4 and the hyperparameters in Eq. 8 is similar to the synthetic data.

**Causal discovery.** Our obtained causal diagram is depicted in Fig. 2(b). As illustrated, Vancomycin exhibits a significant causal influence on White Blood Cell Count and Platelets, which is in accordance with existing clinical research (Rosini et al., 2015; Mohammadi et al., 2017). Furthermore, Fig. 2 demonstrates a close association between the prescription of morphine and Mean Blood Pressure and Platelets, but it does not appear to exert a noticeable influence on White Blood Cell Counts, aligning with the known pharmacological side effects of morphine as reported in Anand et al. (2004); Degrauwe et al. (2019); Simons et al. (2006). Additionally, we observe that Norepinephrine causally affects all blood count parameters, as also found in previous studies (Gader & Cash, 1975; Larsson et al., 1994; Belin et al., 2023).

**Causal Effect Estimation.** We estimate the causal effect of VAN $\rightarrow$ WBC, NE $\rightarrow$ MBP, MS $\rightarrow$ PLT and report the curve of causal effect across dosage in Fig. 3. As shown, vancomycin, used to control bacterial infections in patients with sepsis, initially lowered white blood cell counts and then leveled off because of its bactericidal and anti-inflammatory properties. Besides, Norepinephrine, as a vasopressor, can increase blood pressure. Also, Fig. 3 shows that Morphine has a negative effect on platelet counts. Our findings are consistent with the literature studying the effects of three drugs on blood count indexes (Rosini et al., 2015; Degrauwe et al., 2019; Belin et al., 2023).

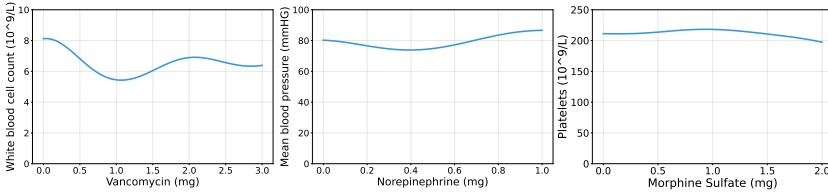

Figure 3: The curve of $\mathbb{E}[Y \,|\, \mathrm{do}(a)]$ with respect to $A$ of WBC (left), MS (middle), and PLT (right).

## 7 CONCLUSIONS AND DISCUSSIONS

In this paper, we introduce a new setting called *multiple treatments and multiple outcomes*, which is a natural extension of the multiple treatments setting to scenarios where multiple outcomes are of interest. Under this new scenario, we show the identifiability of the causal effect if the bipartite graph over treatments and outcomes is not dense enough, which can easily hold. We then employ hypothesis testing for causal discovery to identify proxy variables. With such identified proxies, we estimate the causal effect with a kernel-based doubly robust estimator that is provable to be consistent. We demonstrate the utility on synthetic and real-world data.

**Limitation and Future Works.** Our method requires the proxy variable for hypothesis testing to detect causal edges. Nevertheless, as demonstrated in the experiment presented in Appendix E.2, the type-I error could still be influenced if the selected proxy variables lack sufficient strength. A possible solution to this problem is to select the most appropriate proxy variable according to the Bayes Factor that can be used to control the posterior Type-I error.

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

APPENDIX

# A  PRELIMINARIES

## A.1  NOTATION

In this section, we will define some notations used throughout the proof in the appendix. Moreover, we will introduce other notations in the corresponding subsection.

Table 2: Table of Notations

| Notation | Meaning |
|---|---|
| $\mathbf{X}, \mathbf{U}$ | Covariates and unobserved confounders |
| $\mathbf{A} = [A_i]_{i=1:I}$ | $I$ ($I > 1$) treatments, |
| $\mathbf{Y} = [Y_j]_{j=1:J}$ | $J$ ($J > 1$) outcomes |
| $\mathbf{O}_S = \{O_i | i \in S\}$ | the subset of $\mathbf{O}$ for any variables $O$ that can denote $\mathbf{A}$, $\mathbf{X}$, and $\mathbf{Y}$ |
| $\mathbf{O}_{-S} = \{O_i | i \notin S\}$ | the complementary set of $\mathbf{O}_S$ |
| $\mathrm{do}(\mathbf{a}_S)$ | $\mathrm{do}(\mathbf{A}_S = \mathbf{a}_S)$, intervention variable $\mathbf{A}_S$ with a value of $\mathbf{a}_S$ |
| $Y(\mathbf{a})$ | the potential outcome with $\mathbf{A} = \mathbf{a}$ |
| $\mathbb{E}[\cdot], \mathbb{P}[\cdot]$ | the expectation and the probability distribution of a random variable |
| $\mathcal{P}(X \mid y)$ | $[\mathbb{P}(x_1 \mid y), \cdots, \mathbb{P}(x_k \mid y)]^\top$ |
| $\mathcal{P}(x \mid Y)$ | $[\mathbb{P}(x \mid y_1), \cdots, \mathbb{P}(x \mid y_l)]$ |
| $\mathcal{P}(X \mid Y)$ | $[\mathcal{P}(X|y_1), \cdots, \mathcal{P}(X|y_l)]$ |
| $[I]$ | $\{1, 2, 3, \ldots, n\}$ |
| $|\mathbf{O}_S|$ | the number of subsets of $\mathbf{O}$ is $S$ |
| $W_{S,j}, Z_{S,j}$ | two admissible proxies of $\mathbf{A}_S \rightarrow Y_j$ |
| $\{\mathcal{U}_n\}_{n=1}^N, \{\mathcal{A}_n\}_{n=1}^N$ | measurable partition of $\mathbb{U}$ and $A_{i'}$ |
| $\overline{\mathbf{U}}, \overline{\mathbf{A}}_{i'}$ | discretized version of $\mathbb{U}$ and $A_{i'}$ |
| $q_y$ | $\{\mathbb{P}(Y_j \leq y|a_i)\}_{i=1}^M$ |
| $Q$ | $\left\{\mathcal{P}(\overline{A}_{i'}|a_i)\right\}_{i=1}^M$ |
| $\hat{q}_y, \hat{Q}$ | available estimators of $q_y, Q$ |
| $\mathbf{A}\backslash\mathbf{B}$ | Set subtraction |
| $\mathrm{supp}(A)$ | $\{a \in A : a \neq 0\}$ |
| $\mathrm{TV}(\cdot, \cdot)$ | Total Variation (TV) distance |
| $\|f\|_{\mathrm{TV}}$ | $\frac{1}{2}\|f\|_1$ |
| $\|x\|_1$ | $\sqrt{\sum_i |x_i|}$ |

## A.2  LIPSCHITZNESS

In this section, we introduce the definition of Lipschitz Continuous. Given a metric space $(\mathcal{X}, \rho)$, a function $f\colon X \rightarrow \mathbb{R}$ is L-Lipschitz with respect to the metric $\rho$ if
$$|f(x) - f(x')| \leq L\rho(x, x') \qquad \forall x, x' \in \mathcal{X}.$$

**Definition A.1** (Total variation distance). *The total variation distance between two probability measures $\mathbb{P}$ and $\mathbb{Q}$ on a measurable space $(\Omega, \mathcal{X})$ is defined as*
$$\|\mathbb{P} - \mathbb{Q}\|_{\mathrm{TV}} = \sup_{A \subset \mathcal{X}} |\mathbb{P}(A) - \mathbb{Q}(A)| = \frac{1}{2}\|\mathbb{P} - \mathbb{Q}\|_1.$$

## B  REGULARITY CONDITION

Following Miao et al. (2018); Cui et al. (2020), we use the Picard's Theorem (Carrasco et al., 2007) to characterize the existence of solutions to equations of the first kind by the singular value decomposition of the associated operators.

**Lemma B.1.** *Given Hilbert spaces $H_1$ and $H_2$, a compact operator $K : H_1 \longmapsto H_2$ and its adjoint operator $K' : H_2 \longmapsto H_1$, there exists a singular system $(\lambda_n, \varphi_n, \psi_n)_{n=1}^{+\infty}$ of $K$ with nonzero singular values $\{\lambda_n\}$ and orthogonal sequences $\{\varphi_n \in H_1\}$ and $\{\psi_n \in H_2\}$. Then the equation of the first kind $Kh = \phi$, where $\phi$ be a given function in $H_2$, has solution if and only if*

*1. $\phi \in \mathcal{N}(K')^{\perp}$, where $\mathcal{N}(K') = \{h : K'h = 0\}$ is the null space of the adjoint operator $K'$;*
*2. $\sum_{n=1}^{+\infty} \lambda_n^{-2} |\langle \phi, \psi_n \rangle|^2 < +\infty.$*

In the following two lemmas, we show the existence of bridge functions under the completeness conditions. We replace $W_{S,j}$ and $Z_{S,j}$ with $W$ and $Z$ for brevity

**Lemma B.2.** *Assume Assumption 2 condition 1, Assumption 5 condition 1 and the following conditions for almost all a:*

- $\int \int p(w|z,a)p(z|w,a)\mathrm{d}w\mathrm{d}z < \infty$ and $\int \mathbb{E}^2[Y|z,a]p(z|a)\mathrm{d}z < \infty$;
- $\sum_{n=1}^{\infty} \lambda_{a,n}^{-2}|\langle \mathbb{E}[Y|z,a], \phi_{a,n}\rangle|^2 < \infty.$

*Then there exists function $h \in L_2(W|A = a)$ for almost all a such that*

$$\mathbb{E}[Y|Z, A] = \int h(w, A)\mathrm{d}\mathbb{P}(w|Z, A),$$

**Lemma B.3.** *Assume Assumption 4.3(2) and the following conditions for almost all a:*

- $\int \int p(w|z,a)p(z|w,a)\mathrm{d}w\mathrm{d}z < \infty$ and $\int p^{-2}(a|w)p(w|a)\mathrm{d}w < \infty$;
- $\sum_{n=1}^{\infty} \lambda_{a,n}^{'-2}|\langle p^{-1}(a|w), \phi_{a,n}'\rangle|^2 < \infty.$

*Then there exists function $h \in L_2(Z|A = a)$ for almost all a such that*

$$\mathbb{E}[q(a, Z)|A = a, W] = \frac{1}{p(A = a|W)}$$

## C  IDENTIFICATION

This section comprises proofs for all results presented in sec. 3 and sec. 4, along with additional extensions.

### C.1  PROOF OF LEMMA 4.1

**Lemma 4.1.** *Suppose Assumptions 3.1 and 3.2 hold for $\mathbf{A}_S \subset \mathbf{A}$ and $Y_j \in \mathbf{Y}$. Then, there exist two admissible proxies $(Z_{S,j}, W_{S,j})$ that satisfy Eq. 2, 3.*

*Proof.* We consider a limiting case in which only one of the three assumptions is satisfied.

**1.** If Assumption 3.2 **(i)** is satisfied and Assumption 3.2 **(ii)** and **(iii)** are not satisfied, that implies $|\mathbf{Y}| \geq 3$. Since there exists at least one missing edge from $\mathbf{A}_S$ to $\mathbf{Y}_{-j}$, we can choose $Z_{Sj} \in \{Y \mid \mathbf{A}_{-S} \nrightarrow Y, Y \in \mathbf{Y}_{-j}\}$ and $W_{Sj} \in \mathbf{Y}_{-j} \backslash Z_{Sj}$ as two types of proxies. It follows that there must be no unblocked causal pathway between $Z_{Sj}$ and $Y_j$ conditional on $U, A_i, \mathbf{A}_{-i}\backslash W_{Sj}$, and $Z_{Sj}, A_i \perp W_{Sj} \mid (U, \mathbf{A}_{-i}\backslash W_{Sj})$ since $Z_{Sj}$ only passes $U$ and $\mathbf{A}_{-i}\backslash W_{Sj}$ to $Y_j$. Thus, $Z_{ij}$ and $W_{ij}$ that satisfy the conditional independencies in Conditions 2 and 3 are identifiable.

**2.** If Assumption 3.2 **(ii)** is satisfied and Assumption 3.2 **(i)** and **(iii)** are not satisfied, that implies $|\mathbf{A}_{-S}| \geq 2$. Since there exists at least one missing edge from $\mathbf{A}_{-S}$ to $Y_j$, we can choose $Z_{Sj} \in \{Y \mid \mathbf{A}_{-S} \nrightarrow Y, Y \in \mathbf{Y}_j\}$ and $W_{Sj} \in \mathbf{A}_{-S} \backslash Z_{Sj}$ as two types of proxies.

**3.** If Assumption 3.2 **(iii)** is satisfied and Assumption 3.2 **(i)** and **(ii)** are not satisfied, that implies there is at least one missing edge from $\mathbf{A}_{-S}$ to $\mathbf{Y}_{-j}$. We can choose $(Z_{Sj}, W_{Sj}) \in \{(Y, A) \mid A \nrightarrow Y, A \in \mathbf{A}_{-S}, Y \in \mathbf{Y}_{-j}\}$ as two type of proxies. □

*Remark* C.1. Since we are dealing with a single treatment, we can relax Assumption 3.2. According to Remark 3.3, if the entire bipartite graph between $\mathbf{A}$ and $\mathbf{Y}$ has at most $IJ - 2$ edges, then the proxies are identifiable for each pair $(A_i, Y_j)$. Specifically, if $\deg^+(A_i) = J$ or $\deg^-(Y_j) = I$, where $\deg^-$ and $\deg^+$ refer to the indegree and outdegree of a vertex, then we have $W_{ij}, Z_{ij} \in \{\mathbf{A} \backslash A_i\} \cup \{\mathbf{Y} \backslash Y_j\}$. On the other hand, if $\deg^+(A_i) < J$ and $\deg^-(Y_j) < I$, we can choose $W_{ij}, Z_{ij} \in \mathbf{A} \backslash A_i$ such that $Z_{ij}$ is the vertex that is not connected to $Y_j$ and $W_{ij}$ is a vertex other than these two points.

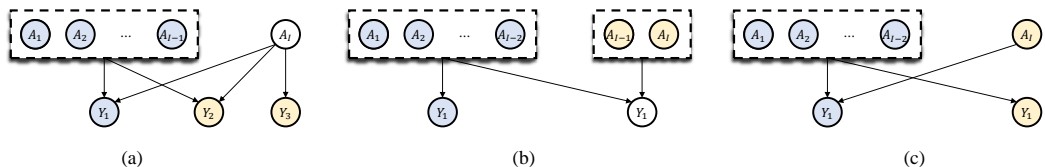

Figure 4: Example diagrams of three proof situations, where we omit confonder $\mathbf{U}$

## C.2 PROOF OF THEOREM 4.4

**Theorem 4.4.** *Suppose assump. 3.1-3.5, 4.3 hold for $\mathbf{A}_S$ and $Y_j$. Then $\mathbb{E}[Y_j | \mathrm{do}(\mathbf{a}_S)]$ is identifiable.*

*Proof.* Based on Lemma 4.1, we can identify two type of proxies $Z_{Sj}$ and $W_{Sj}$, which satisfy the conditional independencies in Eq. 2 and 3, respectively. As a result, under Assumption 3.1-3.5 and Assumption 4.3, we can identify $\mathbb{E}[Y_j \mid \mathrm{do}(\mathbf{a}_S)]$. □

## C.3 PROOF OF THEOREM 4.8

To proceed with our analysis, we now shift our attention to proving Theorem 4.8. To this end, we first present a series of lemmas that will prove useful in our subsequent proof.

**Lemma C.2.** *Assume $z \mapsto \mathbb{P}((X, Y)|Z = z)$ are Lipschitz continuous with respect to Total Variation (TV) distance. Suppose that $\{\mathcal{V}_j\}_{j=1}^J$ is a measurable partition of the support of $Z$, and that $\mathrm{diam}(\mathcal{V}_j) \leq \frac{2\varepsilon}{L}$ for every bin $\mathcal{V}_j$ in the partition. Then, for all $z_0 \in \mathcal{V}_j$, it holds that*

$$\mathrm{TV}\left(\mathbb{P}((X, Y)|Z \in \mathcal{V}_j), \mathbb{P}((X, Y)|Z = z_0)\right) \leq \varepsilon.$$

*Proof.* Consider a Borel measurable function $z \mapsto \mu_z$ defined on a Polish space $S$, where $\mu_z = \mathbb{P}((X, Y)|Z = z)$. Additionally, let $\mathcal{V}_i \subseteq S$ be a Borel set within $S$. We denote $\nu = \mathbb{P}((X, Y)|Z = z_0)$ and $\lambda(Z) = \frac{\mathbb{P}(Z)}{\mathbb{P}(Z \in \mathcal{V}_j)}$ as a probability measure on $\mathcal{V}_j$. Therefore, we obtain the following:

$$\begin{aligned}
&\mathrm{TV}\left(\mathbb{P}((X, Y)|Z \in \mathcal{V}_j), \mathbb{P}((X, Y)|Z = z_0)\right) \\
=&\mathrm{TV}\left(\frac{1}{\mathbb{P}(Z \in \mathcal{V}_j)} \int_{V_i} \mathbb{P}((X, Y)|Z = z) \mathrm{d}\mathbb{P}(z), \mathbb{P}((X, Y)|Z = z_0)\right) \\
=&\mathrm{TV}\left(\int_{\mathcal{V}_i} \mu_z \mathrm{d}\lambda(z), \nu\right) = \mathrm{TV}\left(\int_{\mathcal{V}_i} \mu_z \mathrm{d}\lambda(z), \int_{\mathcal{V}_i} \nu \mathrm{d}\lambda(z)\right) \\
=&\frac{1}{2}\left|\int_{\mathcal{V}_i} \mu_z - \nu \mathrm{d}\lambda(z)\right| \leq \frac{1}{2}\int_{\mathcal{V}_i} |\mu_z - \nu| \mathrm{d}\lambda(z) \\
=&\int_{\mathcal{V}_i} \mathrm{TV}(\mu_z, \nu) \mathrm{d}\lambda(z) \leq \frac{1}{2}L |z - z_0| \leq \varepsilon
\end{aligned}$$

where the last inequation is due to the Lipschitzness of $z \mapsto \mathbb{P}((X, Y)|Z = z)$. □

**Lemma C.3** (Shao (2003)). *Let $X_1, X_2, \cdots$ and $Y$ be random $k$-vectors satisfying $a_n(X_n - c) \to Y$ in distribution, where $c \in \mathbb{R}^k$ and $\{a_n\}$ is a sequence of positive numbers with $\lim_{n \to +\infty} a_n = +\infty$. Let $g$ be a function from $\mathbb{R}^k$ to $\mathbb{R}$. Suppose that $g$ has continuous partial derivatives of order $m > 1$*

*in a neighborhood of c, with all the partial derivatives of order smaller than $m - 1$ vanishing at c, but with the mth-order partial derivatives not all vanishing at c. Then*

$$a_n^m \{g(X_n) - g(c)\} \to \frac{1}{m!} \sum_{i_1=1}^{k} \cdots \sum_{i_m=1}^{k} \frac{\partial^m g}{\partial x_{i_1} \cdots \partial x_{i_m}}\bigg|_{i_m=c} Y_{i_1} \times \cdots \times Y_{i_m} \text{ indistribution,}$$

*where $Y_j$ is the jth component of $Y$.*

**Theorem 4.8.** *Denote $Q_1 := \Sigma_y^{-1/2} Q^\top$, $\Omega_y := I - Q_1 (Q_1^\top Q_1)^{-1} Q_1^\top$. We thus have $\xi_y = \widehat{\Omega}_y \widehat{Q}_1$. We select $a_1, ..., a_M$ with $M > N$. Then under assump. 4.5-4.7, if $\mathbb{H}_0$ is correct, we have $\sqrt{n}\xi_y \xrightarrow{d} N(0, \Omega_y)$, where the rank of $\Omega_y$ is $M - N$ and $n\xi_y^\top \xi_y \to \chi_{M-N}^2$.*

*Proof.* As we mentioned before, we denote $N$ bins $\{\mathcal{U}_n\}_{n=1}^N$ and $\{\mathcal{A}_n\}_{n=1}^N$ as measurable partition of $\mathbf{U}$ and $A_{i'}$ in the hypothesis testing of $\mathbb{H}_0 : A_i \perp Y_j | \mathbf{U}$. Denote $\overline{\mathbf{U}}$ and $\overline{A}_{i'}$ as discretized version of $\mathbf{U}$ and $A_{i'}$ such that $\overline{\mathbf{U}} = k$ iff $\mathbf{U} \in \mathcal{U}_k$. By the definition of conditional probability and $A_i \perp A_{i'} | \mathbf{U}$, we have

$$\mathbb{P}(a_{i'} \in \mathcal{A}_n | a_i) = \sum_{k=1}^N \mathbb{P}(a_{i'} \in \mathcal{A}_n | \mathbf{U} \in \mathcal{U}_k, a_i) \mathbb{P}(\mathbf{U} \in \mathcal{U}_k | a_i) := \mathcal{P}(a_{i'} \in \mathcal{A}_n | \overline{\mathbf{U}}, a_i) \mathcal{P}(\overline{\mathbf{U}} | a_i),$$

$$\mathbb{P}(Y_j \le y | a_i) = \sum_{k=1}^N \mathbb{P}(Y_j \le y | \mathbf{U} \in \mathcal{U}_k, a_i) \mathbb{P}(\mathbf{U} \in \mathcal{U}_k | a_i) := \mathcal{P}(Y_j \le y | \overline{\mathbf{U}}, a_i) \mathcal{P}(\overline{\mathbf{U}} | a_i).$$

For the first equation, we line up all the $\mathcal{A}_n$ in a row,

$$\mathcal{P}(\overline{A}_{i'} | a_i) = \mathcal{P}(\overline{A}_{i'} | \overline{\mathbf{U}}, a_i) \mathcal{P}(\overline{\mathbf{U}} | a_i).$$

By Assumption 4.6, the matrix $\mathcal{P}(\overline{A}_{i'} | \overline{\mathbf{U}}, a_i)$ is invertible, then we obtain

$$\mathbb{P}(Y_j \le y_j | a_i) = \mathcal{P}(Y_j \le y_j | \overline{\mathbf{U}}, a_i) \mathcal{P}(\overline{A}_{i'} | \overline{\mathbf{U}}, a_i)^{-1} \mathcal{P}(\overline{A}_{i'} | a_i)$$

According to Lemma C.2, as long as the partition is fine enough, namely $\text{diam}(\mathcal{U}_k) \le \min\{\varepsilon/L_{A_{i'}}, \varepsilon/L_{A_{i'}|A_i}\}$, we have

$$\text{TV}(\mathbb{P}(Y_j \le y_j | \mathbf{U} \in \mathcal{U}_k), \mathbb{P}(Y_j \le y_j | \mathbf{U} = u)) \le \frac{\varepsilon}{2},$$

$$\text{TV}(\mathbb{P}(Y_j \le y_j | \mathbf{U} \in \mathcal{U}_k, a_i), \mathbb{P}(Y_j \le y_j | \mathbf{U} = u, a_i)) \le \frac{\varepsilon}{2}.$$

If $\mathbb{H}_0$ holds, we have

$$|\mathbb{P}(Y_j \le y_j | \mathbf{U} \in \mathcal{U}_k) - \mathbb{P}(Y_j \le y_j | \mathbf{U} \in \mathcal{U}_k, a_i)|$$
$$\le |\mathbb{P}(Y_j \le y_j | \mathbf{U} \in \mathcal{U}_k) - \mathbb{P}(Y_j \le y_j | a_i, \mathbf{U} = u)|$$
$$+ |\mathbb{P}(Y_j \le y_j | a_i, \mathbf{U} = u) - \mathbb{P}(Y_j \le y_j | \mathbf{U} \in \mathcal{U}_k, a_i)|$$
$$= |\mathbb{P}(Y_j \le y_j | \mathbf{U} \in \mathcal{U}_k) - \mathbb{P}(Y_j \le y_j | \mathbf{U} = u)|$$
$$+ |\mathbb{P}(Y_j \le y_j | \mathbf{U} = u, a_i) - \mathbb{P}(Y_j \le y_j | \mathbf{U} \in \mathcal{U}_k, a_i)|$$
$$\overset{(1)}{\le} 2\text{TV}(\mathbb{P}(Y_j \le y_j | \mathbf{U} \in \mathcal{U}_k), \mathbb{P}(Y_j \le y_j | \mathbf{U} = u))$$
$$+ 2\text{TV}(\mathbb{P}(Y_j \le y_j | \mathbf{U} \in \mathcal{U}_k, a_i), \mathbb{P}(Y_j \le y_j | \mathbf{U} = u, a_i))$$
$$\le \varepsilon$$

where (1) is derived from Def. A.1. Similar, according to Lemma C.2, we have

$$\text{TV}(\mathbb{P}(a_{i'} \in \mathcal{A}_n | \mathbf{U} \in \mathcal{U}_k, a_i), \mathbb{P}(a_{i'} \in \mathcal{A}_n | \mathbf{U} = u, a_i)) \le \varepsilon.$$

Therefore

$$\mathbb{P}(Y_j \le y_j | a_i) = \mathcal{P}(Y_j \le y_j | \overline{\mathbf{U}}, a_i) \mathcal{P}(\overline{A}_{i'} | \overline{\mathbf{U}}, a_i)^{-1} \mathcal{P}(\overline{A}_{i'} | a_i)$$
$$= \left[ \mathcal{P}(Y_j \le y_j | \overline{\mathbf{U}}, a_i) - \mathcal{P}(Y_j \le y_j | \overline{\mathbf{U}}) + \mathcal{P}(Y_j \le y_j | \overline{\mathbf{U}}) \right]$$
$$\cdot \left[ \mathcal{P}(\overline{A}_{i'} | \overline{\mathbf{U}}, a_i)^{-1} - \mathcal{P}(\overline{A}_{i'} | \overline{\mathbf{U}})^{-1} + \mathcal{P}(\overline{A}_{i'} | \overline{\mathbf{U}})^{-1} \right] \mathcal{P}(\overline{A}_{i'} | a_i)$$
$$= \mathcal{P}(Y_j \le y_j | \overline{\mathbf{U}}) \mathcal{P}(\overline{A}_{i'} | \overline{\mathbf{U}})^{-1} \mathcal{P}(\overline{A}_{i'} | a_i) + \Delta \mathcal{P}(\overline{A}_{i'} | a_i)$$

where

$$\Delta = \mathcal{P}\left(Y_j \le y_j | \overline{\mathbf{U}}\right) \left[\mathcal{P}\left(\overline{A}_{i'} | \overline{\mathbf{U}}, a_i\right)^{-1} - \mathcal{P}\left(\overline{A}_{i'} | \overline{\mathbf{U}}\right)^{-1}\right]$$

$$+ \left[\mathcal{P}\left(Y_j \le y_j | \overline{\mathbf{U}}, a_i\right) - \mathcal{P}\left(Y_j \le y_j | \overline{\mathbf{U}}\right)\right] \mathcal{P}\left(\overline{A}_{i'} | \overline{\mathbf{U}}\right)^{-1}$$

$$+ \left[\mathcal{P}\left(Y_j \le y_j | \overline{\mathbf{U}}, a_i\right) - \mathcal{P}\left(Y_j \le y_j | \overline{\mathbf{U}}\right)\right] \left[\mathcal{P}\left(\overline{A}_{i'} | \overline{\mathbf{U}}, a_i\right)^{-1} - \mathcal{P}\left(\overline{A}_{i'} | \overline{\mathbf{U}}\right)^{-1}\right].$$

By $A_i \perp A_{i'} \mid \mathbf{U}$, we have

$$\lim_{\epsilon \to 0} \mathcal{P}\left(\overline{A}_{i'} | \overline{\mathbf{U}}, a_i\right)^{-1} - \mathcal{P}\left(\overline{A}_{i'} | \overline{\mathbf{U}}\right)^{-1} = [0].$$

where $[0]$ denote the zero matrix.

By $\mathbb{H}_0$, we have

$$\lim_{\epsilon \to 0} \mathcal{P}\left(Y_j \le y_j | \overline{\mathbf{U}}, a_i\right) - \mathcal{P}\left(Y_j \le y_j | \overline{\mathbf{U}}\right) = [0].$$

With the above two equations, we have

$$\lim_{\epsilon \to 0} \Delta = [0],$$

$$\lim_{\epsilon \to 0} \mathbb{P}(Y_j \le y_j | a_i) = \mathcal{P}\left(Y_j \le y_j | \overline{\mathbf{U}}\right) \mathcal{P}\left(\overline{A}_{i'} | \overline{\mathbf{U}}\right)^{-1} \mathcal{P}\left(\overline{A}_{i'} | a_i\right).$$

Considering all bins of $\operatorname{supp}(A_i)$, this can be written in the form of transition probability matrix:

$$\lim_{\epsilon \to 0} q_y^T = \mathcal{P}\left(Y_j \le y_j | \overline{\mathbf{U}}\right) \mathcal{P}\left(\overline{A}_{i'} | \overline{\mathbf{U}}\right)^{-1} Q,$$

which means $q_y^T \sim Q$ is linear under $\mathbb{H}_0$.

According to Assumption 4.7, since $\sqrt{n}(\widehat{q}_y - q_y) \xrightarrow{d} N(0, \Sigma_y), \widehat{Q} \xrightarrow{p} Q, \widehat{\Sigma}_y \xrightarrow{p} \Sigma_y$, applying Slutsky's theorem, we have $n^{\frac{1}{2}}(\xi_y - \Omega_y \Sigma_y^{-\frac{1}{2}} q_y) \xrightarrow{D} N(0, \Omega_y \Omega_y^T)$, where $\Omega_y = I - Q_1 \left(Q_1^\top Q_1\right)^{-1} Q_1^\top$. Because $\Omega_y$ is a symmetric, idempotent matrix, we have $n^{1/2}(\xi_y - \Omega_y \Sigma_y^{-1/2} q_y) \xrightarrow{d} N(0, \Omega_y)$. When there are enough bins, $\Omega_y \Sigma_y^{-1/2} q_y \to 0$, which implies that $n^{1/2} \xi_y \xrightarrow{d} N(0, \Omega_y)$.

Because $Q_1$ has rank $N$, $Q_1 \left(Q_1^\top Q_1\right)^{-1} Q_1^\top$ is an idempotent matrix of rank $N$. Hence, $\Omega_y$ is an idempotent matrix of rank $M - N$. For fixed $y$, applying Lemma C.3 with $g(x) = x^\top x$, we have $T_y = g(n^{1/2} \xi_y) \xrightarrow{d} N(0, \Omega_y)^\mathrm{T} N(0, \Omega_y)$.

Because $\Omega_y$ is an idempotent matrix of rank $M - N$, there exists a unitary matrix $V$ such that $V \Omega_y V^\top = \operatorname{diag}(1, \ldots, 1, 0, \ldots, 0)$, a diagonal matrix with $M - N$ eigenvalues equal to one. Thus, $V N(0, \Omega_y) \sim N\{0, \operatorname{diag}(1, \ldots, 1, 0, \ldots, 0)\}$, and $N(0, \Omega_y)^\top N(0, \Omega_y) = \{V N(0, \Omega_y)\}^\top \{V N(0, \Omega_y)\} \sim \chi^2_{M-N}$. Therefore, $n \xi_y^\top \xi_y \xrightarrow{d} \chi^2_{M-N}$. $\square$

Given a significance level of $\alpha$, we can reject the null hypothesis $\mathbb{H}_0$ if $n \xi_y^\top \xi_y$ exceeds the $(1 - \alpha)$th quantile of the $\chi_r^2$ distribution. This ensures that the type I error is no larger than $\alpha$ asymptotically.

## C.4   ADDITIONAL INFORMATION

To satisfy the conditions of Theorem 4.8, we need to estimate $\Sigma_y$ and $Q$. The probability matrix $Q$ can be estimated using empirical probabilities. When considering the covariance matrix, it is important to recognize that each element of the vector $q_y$ corresponds to a distribution function. To estimate it, we can utilize the empirical probability distribution function. The central limit theorem ensures that the estimate converges to a normal distribution as the sample size increases. Alternatively, we have the option to discretize the variable $Y_j$ and divide it into a finite number of bins. Specifically, suppose the bins $\{E_l\}_{l=1}^L$ is a measurable partition of the support of $Y_j$. Then let $q^\top = (q_1^\top, \ldots, q_{L-1}^\top)$; then under $\mathbb{H}_0$, we have

$$q \approx \left\{\mathcal{P}\left(y_j \in E_1 \mid \overline{\mathbf{U}}\right) \mathcal{P}\left(\overline{A}_{i'} | \overline{\mathbf{U}}\right)^{-1}, \ldots, \mathcal{P}\left(y_j \in E_{L-1} \mid \overline{\mathbf{U}}\right) \mathcal{P}\left(\overline{A}_{i'} | \overline{\mathbf{U}}\right)^{-1}\right\} \begin{pmatrix} Q & 0 & 0 \\ \vdots & \vdots & \vdots \\ 0 & 0 & Q \end{pmatrix}.$$

Denote the diagonal matrix on the right hand side as $Q_0$. We can construct a new test statistic $\mathbf{T}$ that aggregates all levels of $Y_j$ by replacing $(\hat{q}_y, \hat{Q})$ with $(\hat{q}, \hat{Q})$ wherever they appear in the construction of $\xi_y$ and $T_y$.

**Corollary C.4.** *Suppose we have available estimators $(\widehat{q}, \widehat{Q}_0)$ that satisfy*

$$\sqrt{n}(\widehat{q} - q) \overset{d}{\to} N(0, \Sigma) \quad \widehat{Q}_0 \overset{p}{\to} Q_0, \widehat{\Sigma} \overset{p}{\to} \Sigma$$

*where $\widehat{\Sigma}$ and $\Sigma$ are positive-definite. Denote $Q_2 := \Sigma^{-1/2} Q_0^\top$, $\Omega := I - Q_2 \left(Q_2^\top Q_2\right)^{-1} Q_2^\top$ and $\xi = \widehat{\Omega}\widehat{Q}_2$. We select $a_1, \cdots, a_M$ with $M > N$. Then under Assumption 4.5, 4.6 and 4.7, if $\mathbb{H}_0$ is correct, we have $\sqrt{n}\xi \overset{d}{\to} N(0, \Omega)$, where the rank of $\Omega$ is $(M - N) \times (L - 1)$ and $n\xi^\top \xi \to \chi^2_{(M-N)(L-1)}$.*

Aggregating all levels of an $M$-category outcome results in a chi-square test with $(M - N)(L - 1)$ degrees of freedom. In the case where $q$ degenerates into distributions of discrete variables, we can estimate it using empirical probabilities, and then estimate the covariance matrix accordingly. Besides, we can also use $q = \{\mathbb{E}(Y_j \mid a_i), \ldots, \mathbb{E}(Y_j \mid a_i)\}^\top$ in construction of the test statistic and perform the test on the mean scale.

When there are also observed confounders $\mathbf{X}$, the above proof also holds. We just need to notice

$$\mathbb{P}\left(a_{i'} \in \mathcal{A}_n | a_i, \mathbf{x}\right) = \mathcal{P}\left(a_{i'} \in \mathcal{A}_n | \overline{\mathbf{U}}, a_i, \mathbf{x}\right) \mathcal{P}\left(\overline{\mathbf{U}} | a_i, \mathbf{x}\right),$$
$$\mathbb{P}\left(Y_j \leq y | a_i, \mathbf{x}\right) = \mathcal{P}\left(Y_j \leq y | \overline{\mathbf{U}}, a_i, \mathbf{x}\right) \mathcal{P}\left(\overline{\mathbf{U}} | a_i, \mathbf{x}\right).$$

The rest of the proofs are similar.

In sec. 4, the hypothesis testing is presented under the assumption that the random variables $\mathbf{U}$ and $A_{i'}$ are bounded. However, it is important to note that our hypothesis test can also be applied when the random variables are unbounded. This is due to the fact that when the obtained bound is sufficiently large, the probability of falling in the tail of the distribution becomes arbitrarily small. Specifically, we divide the domain of $\mathbf{U}$ into $\Omega_{\mathbf{U}} := \{|\mathbf{U}| \leq T\} \cup \{|\mathbf{U}| > T\}$, and further partition $\{|U| \leq T\} := \cup_n^N \mathcal{U}_n$.

$$\mathbb{P}\left(Y_j \leq y_j | a_i\right) = \sum_{k=1}^N \mathbb{P}\left(Y_j \leq y_j, \mathbf{U} \in \mathcal{U}_k | a_i\right) + \mathbb{P}\left(Y_j \leq y_j, |\mathbf{U}| > T | a_i\right)$$

$$= \sum_{k=1}^N \mathbb{P}\left(Y_j \leq y_j | \mathbf{U} \in \mathcal{U}_k, a_i\right) \mathbb{P}\left(\mathbf{U} \in \mathcal{U}_k | a_i\right)$$
$$+ \mathbb{P}\left(Y_j \leq y_j | |\mathbf{U}| > T, a_i\right) \mathbb{P}\left(|\mathbf{U}| > T | a_i\right)$$

$$= \sum_{k=1}^N \mathbb{P}\left(Y_j \leq y_j | \mathbf{U} \in \mathcal{U}_k, a_i\right) \mathbb{P}\left(\mathbf{U} \in \mathcal{U}_k | a_i\right) + \varepsilon_T \mathbb{P}\left(Y_j \leq y_j | |\mathbf{U}| > T, a_i\right)$$

$$\mathbb{P}(a_{i'} \in \mathcal{A}_n | a_i) = \sum_{k=1}^N \mathbb{P}\left(a_{i'} \in \mathcal{A}_n, \mathbf{U} \in \mathcal{U}_k | a_i\right) + \mathbb{P}\left(a_{i'} \in \mathcal{A}_n, |\mathbf{u}| > T | a_i\right)$$

$$= \sum_{k=1}^N \mathbb{P}\left(a_{i'} \in \mathcal{A}_n | \mathbf{U} \in \mathcal{U}_k, a_i\right) \mathbb{P}\left(\mathbf{U} \in \mathcal{U}_k | a_i\right)$$
$$+ \mathbb{P}\left(a_{i'} \in \mathcal{A}_n | |\mathbf{u}| > T, a_i\right) \mathbb{P}\left(|\mathbf{u}| > T | a_i\right)$$

$$= \sum_{k=1}^N \mathbb{P}\left(a_{i'} \in \mathcal{A}_n | \mathbf{U} \in \mathcal{U}_k, a_i\right) \mathbb{P}\left(\mathbf{U} \in \mathcal{U}_k | a_i\right) + \varepsilon_T \mathbb{P}\left(a_{i'} \in \mathcal{A}_n | |\mathbf{u}| > T, a_i\right)$$

Upon conducting our analysis, we have discovered that when $T$ is large enough, the probability of falling the tail region becomes arbitrarily small. Moreover, this item has nothing to do with $a_i$, so it will not affect our analysis of whether it is linear.

## C.5 NULL-TV LIPSCHITZNESS

To start, we'll provide some specific examples of distributions that fall into different Lipschitzness classes.

**Example C.5.** *Suppose $(X, Z) \sim N(\mu_1, \mu_2, \sigma_1^2, \sigma_2^2, \rho)$, where $\rho$ is the correlation between $X$ and $Z$, then $z \mapsto \mathbb{P}(X|Z = z)$ be NULL-TV Lipschitzness.*

*Proof.* Easy to obtain the conditional distribution of $X$ given $z$ is given by the normal distribution $X \mid z \sim N(\mu_3, \sigma_3^2)$, where $\mu_3 = \mu_1 + \rho \frac{\sigma_1}{\sigma_2}(z - \mu_2)$ and $\sigma_3^2 = \sigma_1^2 (1 - \rho^2)$. We would like to prove that the function $z \mapsto \mathbb{P}(X|Z = z)$ is NULL-TV Lipschitz, which means that it has a Lipschitz constant. To do so, we calculate the total variation distance between $\mathbb{P}(X|Z = z)$ and $\mathbb{P}(X|Z = z')$.

$$
\begin{aligned}
\text{TV}\left(\mathbb{P}(X|Z = z), \mathbb{P}(X|Z = z')\right) &= \frac{1}{2} \int p(x \mid z) - p(x \mid z') \, \mathrm{d}x \\
&= \frac{1}{2} \int \frac{\partial p(x|\xi)}{\partial z} (z - z') \, \mathrm{d}x \\
&= \frac{1}{2} |z - z'| \int \frac{\partial p(x|\xi)}{\partial z} \mathrm{d}x
\end{aligned}
$$

Thus we find that the total variation distance is bounded by $L |z - z'|$, where $L$ is a Lipschitz constant that depends on the partial derivative of the probability density function with respect to $z$. Observe that $\frac{\partial p(x|\xi)}{\partial z}$ is a function of the form $a(x + b) \exp(-c(x + d)^2)$, which is absolute integrable. we can conclude that $L$ is finite, which implies that $z \mapsto \mathbb{P}(X|Z = z)$ is indeed NULL-TV Lipschitz. $\square$

In fact, Neykov et al. (2021) demonstrated that if the log-conditional density is sufficiently smooth, the resulting distribution belongs to the TV Lipschitzness class. A wide range of exponential family distributions possess a smooth log-conditional distribution, which satisfies Assumption 4.5.

**Example C.6.** *Suppose that $g(x, y, z) : [0, 1]^3 \mapsto [-M, M]$ is a bounded $L$-Lipschitz function, i.e., $|g(x, y, z) - g(x', y', z')| \le L(|x - x'| + |y - y'| + |z - z'|)$. Take $\mathbb{P}(X, Y, Z) \propto \exp(g(x, y, z))$. Then*

$$
p_{X,Y|Z}(x, y|z) = \frac{\exp(g(x, y, z))}{\int_{[0,1]^2} \exp(g(x, y, z)) \mathrm{d}x \mathrm{d}y},
$$

*be NULL-TV Lipschitz which Lipschitz constant is $e^{2L} - 1$.*

Besides, in Bayesian networks, it is often useful to determine the total variation distance between the joint distributions of several random variables. If each of these distributions is log-Lipschitz continuous, then their product is guaranteed to be TV-Lipschitz continuous. This result is particularly useful in Bayesian networks, where it can be used to simplify the analysis of the propagation of probabilities between nodes. Refer to Literature Honorio (2012) for details.

Recently, Dolera & Mainini (2020) provide general conditions for getting a global form of Lipschitz continuity for dominating probability kernels, sharing the common form:

$$
\pi(B|x) := \int_B g(x, \theta) \pi(\mathrm{d}\theta) \qquad \forall B \in \mathcal{F}.
$$

For Exponential models, namely

$$
f(x|\theta) = e^{\Phi(x,\theta)} h(x), \quad g(x, \theta) = \frac{f(x|\theta)}{\rho(x)} = \frac{e^{\Phi(x,\theta)}}{\int_\Theta e^{\Phi(x,\tau)} \pi(\mathrm{d}\tau)}
$$

for some measurable functions $h : \mathbf{X} \to (0, +\infty)$ and $\Phi : \mathbf{X} \times \Theta \to \mathbb{R}$. Here, $\pi$ denotes the prior probability measure. If $\theta \mapsto \nabla_x \Phi(x, \cdot)$ is Lipschitz for any $x \in \mathbf{X}$, then we have NULL-TV Lipschitz constant

$$
L := \operatorname*{ess\,sup}_{x \in \mathbf{X}} \left(\mathcal{C}[g(x, \cdot) \pi)]\right)^2 \left(\int_\Theta |\nabla_\theta \Psi(x, \theta)|^2 g(x, \theta) \pi(\mathrm{d}\theta)\right)^{\frac{1}{2}} < +\infty
$$

In the field of causal inference, recent works by Farokhi (2023); Guo et al. (2022) have employed similar assumptions, leading to partial identification of causal effects in the presence of noisy covariates.

# D  ESTIMATION

With proxies $W$ and $Z$, we should solve the nuisance functions $h$, $q$ in Fredholm integral Equations equation 5. After estimating $h$ and $q$, we employ Colangelo & Lee (2020); Wu et al. (2023) to estimate the causal effect:

$$\mathbb{E}[Y \,|\, \mathrm{do}(a)] \approx \mathbb{E}_n[K_{h_{\mathrm{bw}}}(A - a)\, q(a, Z)(Y - h(a, W)) + h(a, W))], \tag{9}$$

where the indicator function $\mathbb{I}(A = a)$ in the doubly robust estimator for binary treatments Colangelo & Lee (2020) is replaced with the kernel function $K_{h_{\mathrm{bw}}}(a_i - a) = 1/h_{\mathrm{bw}}K\left((a_i - a)/h_{\mathrm{bw}}\right)$ ($h_{\mathrm{bw}} > 0$ is the bandwidth), as a smooth approximation to make the estimation for continuous treatments feasible. In Wu et al. (2023), this estimator coupled with Eq. 6 was shown to enjoy the optimal convergence rate with $h_{\mathrm{bw}} = O(n^{-1/5})$. Following Wu et al. (2023), we need to some assumptions about kernel function.

**Assumption D.1.** *The second-order symmetric kernel function $K(\cdot)$ is bounded differentiable, i.e., $\int k(u)\mathrm{d}u = 1, \int uk(u)\mathrm{d}u = 0, \kappa_2(K) = \int u^2 k(u)\mathrm{d}u < \infty$. We define $\Omega_2^{(i)}(K) = \int (k^{(i)}(u))^2\mathrm{d}u$.*

Assump. D.1 adheres to the conventional norms within the domain of nonparametric kernel estimation and maintains its validity across widely adopted kernel functions, including but not limited to the Epanechnikov and Gaussian kernels

**Theorem D.2.** *Under assump. 3.1, 3.2 and 4.3 and D.1, suppose $\|\hat{h} - h\|_2 = o(1)$, $\|\hat{q} - q\|_2 = o(1)$ and $\|\hat{h} - h\|_2\|\hat{q} - q\|_2 = o((nh_{\mathrm{bw}})^{-1/2})$, $nh_{\mathrm{bw}}^5 = O(1)$, $nh_{\mathrm{bw}} \to \infty$, $h_0(a, w, x), p(a, z|w, x)$ and $p(a, w|z, x)$ are twice continuously differentiable wrt $a$ as well as $h_0, q_0, \hat{h}, \hat{q}$ are uniformly bounded. Then for any $a$, we have the following for the bias and variance of the PKDR estimator given Eq. 8:*

$$\mathrm{Bias}(\hat{\beta}(a)) := \mathbb{E}[\hat{\beta}(a)] - \beta(a) = \frac{h_{\mathrm{bw}}^2}{2}\kappa_2(K)B + o((nh_{\mathrm{bw}})^{-1/2}), \mathrm{Var}[\hat{\beta}(a)] = \frac{\Omega_2(K)}{nh_{\mathrm{bw}}}(V + o(1)),$$

*where $B = \mathbb{E}[q_0(a, Z)[\frac{\partial}{\partial A}h_0(a, W)\frac{\partial}{\partial A}p(a, W|Z) + \frac{1}{2}(\frac{\partial^2}{\partial A^2}h_0(a, W))]], V = \mathbb{E}[\mathbb{I}(A = a)q_0(a, Z)^2(Y - h_0(a, W))^2]$.*

The proof of the Thm. D.2 is detailed in Wu et al. (2023).

# E  EXPERIMENTS

In this appendix, we provide more details of two experiments of sec. 6.1. Besides, we also conduct a new experiment to illustrate the limitation of hypothesis testing in Sec. E.2.

## E.1  SYNTHETIC STUDY OF SEC. 6.1

We present the results of hypothesis testing and offer implementation details of causal estimation in the context of synthetic data.

### E.1.1  HYPOTHESIS TESTING FOR CAUSAL DISCOVERY

**Data generation.** We describe the data generating mechanism of section 6.1. We consider five treatments $\{A_i\}_{i=1}^5$, four outcomes $\{Y_i\}_{i=1}^4$, and one unobserved confounder $U$. We use the following structural equations: $U \sim \mathrm{Uniform}(-1, 1)$, $A_i = g_i(U) + \epsilon_i$, with $g_i$ randomly chosen from

$\{linear, tanh, sin, cos, sigmoid\}$ and $\epsilon_i \sim N(0,1)$ and $\mathbf{Y}$ is a non-linear structure. Specifically,

$$\begin{cases} A_1 = 0.5 \times (U + 5) + \varepsilon_1 \\ A_2 = 0.5 \times (\tanh(U) + 3) + \varepsilon_2 \\ A_3 = 0.5 \times (\sin(\frac{\pi}{8}U) + 3) + \varepsilon_3 \\ A_4 = 0.5 \times (\frac{1}{1.0 + \exp(U) + 3}) + \varepsilon_4 \\ A_5 = 0.5 \times (\cos(\frac{\pi}{8}U) + 3) + \varepsilon_5 \end{cases}$$

$$\begin{cases} Y_1 = 2\sin(1.4A_1 + 2A_3^2) \\ \qquad +0.5(A_2 + A_4^2 + A_5) + A_3^3 + U + \epsilon_1 \\ Y_2 = -2\cos(1.8A_2) + 1.5A_4^2 + U + \epsilon_2 \\ Y_3 = 0.7A_3^2 + 1.2A_4 + U + \epsilon_3 \\ Y_4 = 1.6e^{-A_1 + 1} + 2.3A_5^2 + U + \epsilon_4 \end{cases}$$

where $\epsilon_i \sim N(0,1)$.          where $\epsilon_i \sim N(0,1)$.

**Independence test: i) Fisher** that Fisher conditional independence test; **ii) POP** that The Promises of Parallel Outcomes; **iii) Ours** that proxy based approach.

**Implementation details.** For hypothesis testing, we set the significant level $\alpha$ as 0.05, and the bin numbers for discretization of $A, W, Y$ as $I := |A| = 15, K := |W| = 8, L := |Y| = 5$, respectively. The asymptotic estimators in Eq. 4 are obtained by empirical probability mass functions. The sample size is set to $\{400; 600; 800; 1,000; 1,200; 1,400; 1,600\}$. To remove the effect of randomness, we repeat for 50 replications.

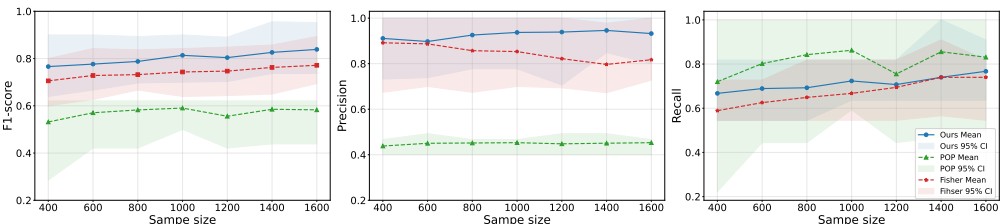

Figure 5: Performance of our method and baselines under the setting of multi-treatments and multi-outcomes.

**Results.** As Fig. 5, we present the performance of our method and the baselines in the context of multiple treatments and outcomes. Notably, our method consistently outperforms all other alternatives across most metrics. It is evident that the POP method which operates within a similar multi-outcome setting, exhibits significantly poor performance due to its dependence on the assumption of linear structural equations. In comparison to Fisher testing, our approach not only demonstrates superior performance but also exhibits greater stability, as it effectively leverages information from proxy variables associated with other treatments.

**Influence of bin numbers.** Fig. 6 shows the $F_1$ score, precision, and recall of our method with bin numbers varying from 2 to 11. As observed, precision improves as we increase the number of bins, up to a point where it reaches its peak at 9 bins, beyond which further increments do not yield significant improvements.

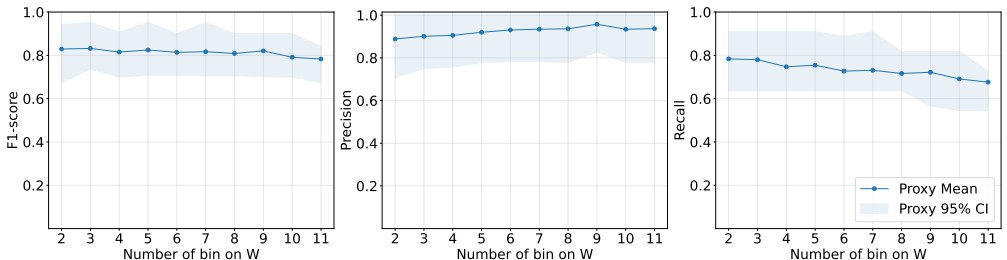

Figure 6: $F_1$ score, precision, and recall of our method under different bin numbers.

### E.1.2   CAUSAL EFFECT ESTIMATION IN SEC. 6.1

We introduce implementation details of the causal effect estimation.

**Implementation details.** For $A_3 \rightarrow Y_1$, $A_2 \rightarrow Y_2$ and $(A_1, A_3) \rightarrow Y_1$, we select two proxies: $Z = Y_3$ and $W = A_5$. For $(A_1, A_5) \rightarrow Y_4$, we select two proxies: $Z = Y_2$ and $W = A_3$. For the PMMR method, we use the Gaussian kernel where the bandwidth is determined by the median trick. Specifically, we set $\gamma^{-1} = \text{median}\{\|x_i - x_j\|_2^2\}_{i<j \in I}$ for indices subset $I \subset \{1, \ldots, n\}$. Regarding the regularization parameters, denoted as $\lambda_h$ and $\lambda_q$, we select them according to the Tab. 3.

Table 3: Hyperparameters for and PMMR models.

|  | $\lambda_{h_1}$ | $\lambda_{h_2}$ | $\lambda_{q_1}$ | $\lambda_{q_2}$ |
|---|---|---|---|---|
| $A_3 \rightarrow Y_1$ | 0.05 | 0.05 | 0.20 | 0.20 |
| $A_2 \rightarrow Y_2$ | 0.20 | 0.20 | 1.00 | 1.00 |
| $(A_1, A_3) \rightarrow Y_1$ | 0.20 | 0.20 | 1.00 | 1.00 |
| $(A_1, A_5) \rightarrow Y_4$ | 0.20 | 0.20 | 0.20 | 0.20 |

For density estimation, we also choose the Gaussian kernel. For bandwidth, we employ three-fold cross-validation, where the bandwidth is chosen as 20 values uniformly distributed in logarithmic space, ranging from $10^{-0.1}$ to 1.

### E.1.3 VISUALIZING DISTRIBUTION OF TREATMENTS AND OUTCOMES

As shown in Fig. 7, all treatments tend to be distributed according to a normal distribution, primarily influenced by the noise in the data. Conversely, the four observed outcomes manifest a predominantly long-tailed distribution.

### E.2 LIMITATIONS ABOUT HYPOTHESIS TESTING IN SEC. 7

In the conclusion section, we mention that the testing may suffer from large type-I errors in causal relations identification if the strength of the proxy variable is not strong enough. In order to gain a more comprehensive understanding of this issue, we measure the effect of varying proxy variable strength on the hypothesis testing results.

**Data generation.** Here we conduct two experiments, whose structural equations are listed as follows. For each experiment, we generate 1,000 samples. To explore the impact of proxy variable strength on the hypothesis testing, we set the coefficient of the proxy variables $\beta$ change from 0.1, 1, 10, 20, 50, and 100.

Experiment (I)

$$\begin{cases} U \sim N(0, 1), \\ A = U + \varepsilon_1, \\ W = \beta \times U + \varepsilon_2, \\ Y = \begin{cases} A + U + \varepsilon_3, & A \rightarrow Y \\ U + \varepsilon_3, & A \nrightarrow Y. \end{cases} \end{cases}$$

where $\epsilon_i \sim N(0, 1)$.

Experiment (II)

$$\begin{cases} U \sim N(0, 1), \\ A = U + \varepsilon_1, \\ W = \beta \times \tanh(U) + \varepsilon_2, \\ Y = \begin{cases} A + U + \varepsilon_3, & A \rightarrow Y \\ U + \varepsilon_3, & A \nrightarrow Y. \end{cases} \end{cases}$$

where $\epsilon_i \sim N(0, 1)$.

**Implementation details.** For hypothesis testing, we set the significant level $\alpha$ as 0.05, and the bin numbers for discretization of $A, W, Y$ as $I := |A| = 15, K := |W| = 8, L := |Y| = 5$. The asymptotic estimators in Eq. 4 are obtained by empirical probability mass functions. To remove the effect of randomness, we repeat for 50 replications.

**Results.** For $A \rightarrow Y$, we report the type-II error to see if we successfully detect this edge, while for $A \nrightarrow Y$, we report the type-I error to see whether we falsely reject the null hypothesis. According to Tab. 4, we find that when $A \nrightarrow Y$, we get a larger type-I error as the strength decreases. This is because in scenarios where $A \nrightarrow Y$, as the proxy strength diminishes, its ability to explain the variable $U$ decreases. Consequently, it becomes challenging to utilize them in hypothesis testing. Moreover, we find that non-linear functions can make it more difficult to infer independent relationships, thus requiring strong proxy strength.

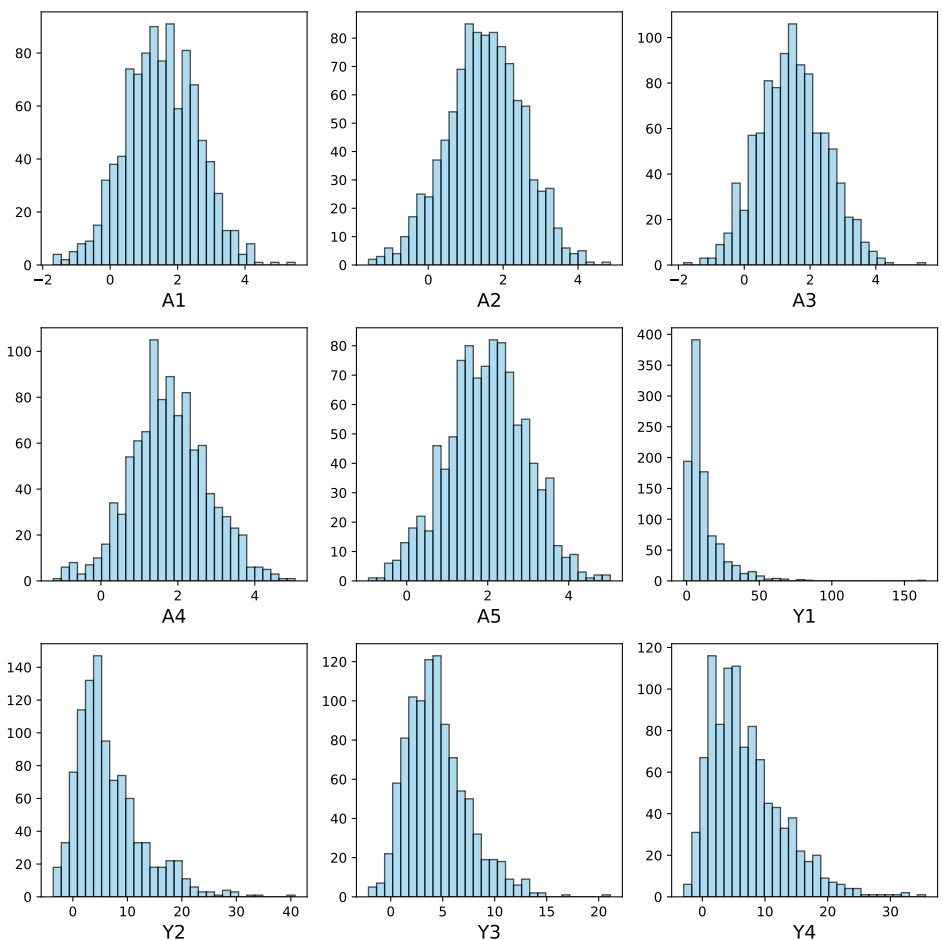

Figure 7: Histograms of Synthetic dataset; sample size=1000.

Table 4: The experimental results in the presence of proxy variables of different strengths. For $A \to Y$, we report the Type-II error; for $A \not\to Y$, we report the Type-I error.

| Proxy strength | Linear | | Nonlinear | |
| :---: | :---: | :---: | :---: | :---: |
| | $A \to Y$ (Type-II) | $A \not\to Y$ (Type-I) | $A \to Y$ (Type-II) | $A \not\to Y$ (Type-I) |
| $\beta = 0.1$ | 0.00 | 1.00 | 0.00 | 1.00 |
| $\beta = 1$ | 0.00 | 0.24 | 0.00 | 0.58 |
| $\beta = 10$ | 0.02 | 0.04 | 0.02 | 0.12 |
| $\beta = 20$ | 0.00 | 0.06 | 0.00 | 0.01 |
| $\beta = 50$ | 0.00 | 0.12 | 0.00 | 0.04 |
| $\beta = 100$ | 0.02 | 0.04 | 0.02 | 0.02 |

### E.3 REAL-WORD STUDY OF SEC. 6.2

We introduce more details for the experiment in Sec. 6.2, including hypothesis testing, implementation details of the causal effect estimation, and visualization of treatments and outcomes' distributions.

**Hypothesis Testing of Baselines** We present the estimated causal graph by two baseline methods, Fisher's independence test and POP. The learned DAGs are shown in Fig 8.

As shown, the DAGs estimated by both methods do not align well with established medical knowledge. For instance, previous research, such as the study by Gader & Cash (1975), has demonstrated that Norepinephrine indeed leads to an increase in blood pressure however the Fisher method fails to detect this relation. Furthermore, both methods find that all treatments have no significant effect on platelets, which is different from existing studies (Belin et al., 2023).

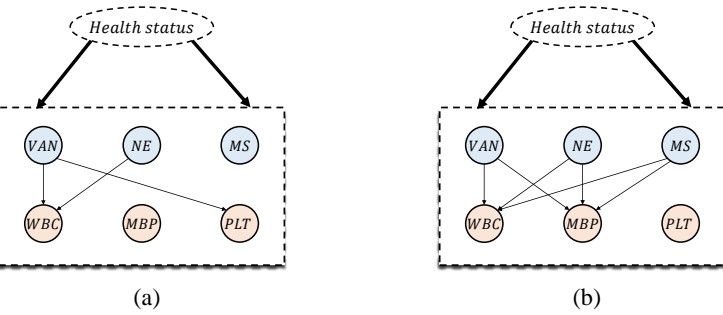

Figure 8: The causal graph over treatments (blue) and outcomes (yellow) of (a) Fisher (b) POP.

**Implementation details of causal effect estimation.** For NE $\to$ MBP, we select two proxies: $Z = \text{WBC}$ and $W = \text{MS}$ and $X = \text{NE}$. For VAN $\to$ WBC, we select two proxies: $Z = \text{PLT}$ and $W = \text{MBP}$. For MS $\to$ PLT, we select two proxies: $Z = \text{MBP}$ and $W = \text{VAN}$. For PMMR method, we use the Gaussian kernel where the bandwidth is determined by the median trick. Regarding the regularization parameters, denoted as $\lambda_h$ and $\lambda_q$, we select it according to the Tab. 5.

Table 5: Hyperparameters for PMMR models in mimic dataset

|  | $\lambda_{h_1}$ | $\lambda_{h_2}$ | $\lambda_{q_1}$ | $\lambda_{q_2}$ |
|---|---|---|---|---|
| VAN $\to$ WBC | 0.2 | 0.2 | 0.1 | 0.1 |
| NE $\to$ MBP | 0.2 | 0.2 | 0.2 | 0.2 |
| MS $\to$ PLT | 0.2 | 0.2 | 0.2 | 0.2 |

**Distribution of Treatments and Outcomes.** As Fig. 9, we present the histograms for the treatments and outcomes observed in the MIMIC. It is important to note that normal reference ranges for blood indicators in healthy individuals are as follows: White Blood Cell Count (4-10), Mean Blood Pressure (70-105), and Platelets (100-300).

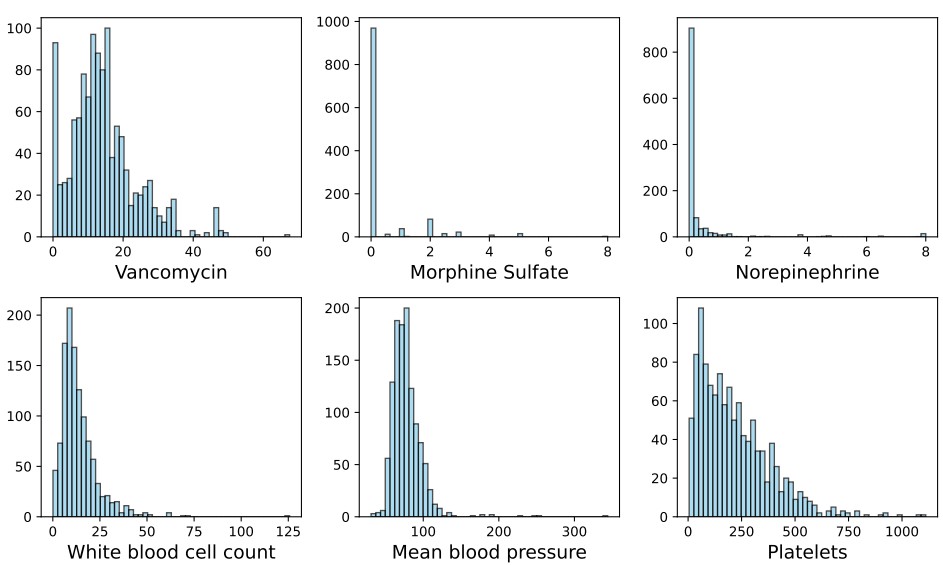

Figure 9: Histograms of treatments and outcomes in MIMIC dataset; sample size=1165.

