# OpenReview forum: "The Blessings of Multiple Treatments and Outcomes in Treatment Effect Estimation"
_ICLR.cc/2024/Conference — ICLR 2024 Conference Withdrawn Submission_

### Official Review · Reviewer_4KmT · 2023-10-30

**Soundness:** 3 good
**Presentation:** 3 good
**Contribution:** 2 fair
**Rating:** 5
**Confidence:** 3

**Summary:**

This paper proposes a method for estimating treatment effects in the presence of multiple outcomes and multiple treatments and shared unobserved confounders. The idea extends the setting in treatment effect estimation from multiple treatments, by Wang and Blei 2019. The proposed estimation algorithm involves a two steps paradigm: first identifying the proxy variables using causal structure learning, and then conduct inference with existing estimators. The methodology is further supplemented by identification results, along with synthetic studies and an experiment using the MIMIC III dataset.

**Strengths:**

1. The motivation of causal effect estimation in presence of multiple treatments and outcomes, and especially utilize the multiple treatments/outcomes as proxies is novel and significant.
2. The identifiability results seems solid and does not require pre-specified proxies. Although I have not checked the technical proof details.

**Weaknesses:**

1. The experiment evaluation is weak when compared to other causal inference methods submitted to ICLR or similar AI/ML conferences. None of the benchmark neural network-based treatment effect estimation methods have been included in the baseline, e.g., CFR, DR-CFR, CEVAE, TEDVAE, etc. Most of these mentioned methods have been published in AAAI/ICLR/NeurIPS. They are not designed for multiple outcomes/treatments, but comparing with these methods can give a clear practical implication to the proposed method.
2. This paper is written towards an audience of applied statistics researchers, and is more suited for a statistical journal such as Biometrika. Although the reviewer acknowledge the importance of rigorous discussing identifiability, the methodology may not be very relevant to the broader community of ICLR. For example, it does not discuss neural network or representation learning at all.
3. The method requires an additional step of causal discovery for learning the causal structure.
4. The two-step algorithm design may be suboptimal, especially when the estimation in the second step cruciallly depends on the causal discovery results in the first step.

**Questions:**

What if (i) of Assumption 3.1 is not satisfied? Or specifically, does $\mathbf{U}$ has to be the same for all outcomes? Can a subset of $\mathbf{U}$ affect some outcomes, and other subsets affect other outcomes?

---

### Official Review · Reviewer_9yEQ · 2023-10-30

**Soundness:** 3 good
**Presentation:** 3 good
**Contribution:** 3 good
**Rating:** 6
**Confidence:** 5

**Summary:**

This paper considers a setting with multiple treatments and multiple outcomes. They show that parallel studies of multiple outcomes involved in this setting can assist each other in causal identification, in the sense that one can exploit other treatments and outcomes as proxies for each treatment effect under study. They proceed with a causal discovery method that can effectively identify such proxies for causal estimation.

**Strengths:**

The work is pretty sound. The idea of using multiple outcomes and treatments as proxies is interesting.

**Weaknesses:**

My main concern is about the inference on selected proxies. Have the authors considered post selection problem as the convergence results considered in the Appendix seem to assume a valid Z, W and ignore the selection step?

**Questions:**

If there are multiple Z and W satisfy (2) & (3), how shall I pick?

Is Assumption 4.6 a strong Assumption?

---

### Official Review · Reviewer_928E · 2023-10-31

**Soundness:** 3 good
**Presentation:** 3 good
**Contribution:** 3 good
**Rating:** 5
**Confidence:** 4

**Summary:**

The paper "The Blessings of Multiple Treatments and Outcomes in Treatment Effect Estimation" proposes to extend existing results relying on multiple treatments or multiple outcomes to provide additional adjustment and better enforce the ignorability assumption in the presence of unobserved confounders. The authors propose a theoretical proof to demonstrate the identifiability of the average treatment effect (ATE) and an algorithmic strategy to estimate the ATE in that setting. Finally, experiments on simulated and real data are conducted to illustrate the performances of the new proposed approach.

**Strengths:**

The article is well written, and technically sound. The proposed approach is interesting, with great potential for practical application. The previous literature is well integrated, and the paper's novelty is well explicited.

**Weaknesses:**

1. The proposed extension is novel, and it is very valuable for further practical application to have a sound technical ground, however, the idea is not revolutionary, but rather a natural extension of existing work.

2. The deconfounder approach (Wang and Blei) has faced extensive criticism, with a clear description of settings where this approach is relevant, and of settings where it fails. A similarly clear description of the application field is needed for more applied practitioners.

3. The final "algorithm" is very hard to understand from the sole reading of the article, as many parts of the implemented strategy are just referred to as citations. It would greatly improve the usefulness of the article to provide a clear outline that could be implemented (probably in Supplementary materials). A graphical illustration of the steps (causal discovery, and then estimation) would also really help the readability of the article, especially for applied researchers.

4. The authors should be commended for providing source code for their work, however, it does not seem to be oriented towards being an actual package that could be installed, and used easily by anyone (no mention of the code in the article, maybe in a reproducibility section as suggested in the ICLR author guide, no documentation, such as a README or a tutorial)

**Questions:**

main questions
5. The experiments should help clarifying when to use or not use the proposed approach, but the only simulated setting concerns a case where both multiple treatments and outcomes can be leveraged to deconfound. How does the proposed approach perform compared to the deconfounder and POP when only multiple treatments or multiple outcomes can be used?

6. Can you provide the results of all methods to the real MIMIC data application? It would provide insights about the differences in the final results in a real-world application, in addition to simulated data designed to illustrate the benefit of the proposed approach in a favorable case.

7. What would be a way to provide an estimation uncertainty for the ATE in the proposed method? Beyond the estimate, that is an interesting metric to report on real data, for the proposed approach and the baselines.

8. What is the relative required dimensionality of the treatment and outcome compared to the unobserved confounders dimension?


details
9. duplicated bibliography item "Yifan Cui, Hongming Pu, Xu Shi, Wang Miao, and Eric Tchetgen Tchetgen. Semiparametric proximal causal inference."
10. the associated code should be mentioned (maybe in a reproducibility section as suggested in the ICLR author guide)
11. the associated code is not documented. A readme, a tutorial to use etc should be included
12. the proposed method does not have a name
13. PMMR meaning is not defined

---

### Official Review · Reviewer_RbZe · 2023-11-01

**Soundness:** 2 fair
**Presentation:** 2 fair
**Contribution:** 2 fair
**Rating:** 5
**Confidence:** 3

**Summary:**

The authors extend earlier work on multiple treatments and single outcomes to the more general setting of multiple treatments and outcomes. They use proximal causal learning to show causal identifiability and use a hypothesis testing approach to find proxies which can then be used to estimate the causal effect. They demonstrate their approach on a synthetic and a clinical dataset.

**Strengths:**

The authors theoretically motivate their approach and demonstrate it works on two datasets.

The paper is generally easy to follow.

**Weaknesses:**

The experiments are modest in size with relatively simple causal relationships. It's not clear if their approach would work with more complex datasets and relationships. For instance, as I understand the authors assume that there is no interaction between treatments and outcomes. To what extent would this approach still work if this assumption was violated?

**Questions:**

How would the proposed method scale to larger datasets with more complex causal relationships?